

# Global nutrient cycling by commercially-targeted marine fish

Priscilla Le Mézo[1,2], Jérôme Guiet[3], Kim Scherrer[1], Daniele Bianchi[3], and Eric Galbraith[1,4,5]

[1]Institut de Ciencia i Technologia Ambientals (ICTA), Universitat Autonoma de Barcelona (UAB), Barcelona, Spain
[2]Laboratoire de Météorologie Dynamique, ENS Ulm, Paris, France
[3]Atmospheric and Oceanic Sciences, University of California, Los Angeles, CA, United States
[4]Catalan Institution for Research and Advanced Studies (ICREA), Barcelona, Spain
[5]Earth and Planetary Sciences, McGill University, Montreal, QC, Canada

**Correspondence:** Priscilla Le Mézo (priscilla.le-mezo@lmd.ens.fr)

**Abstract.** Throughout the course of their lives fish ingest food containing essential elements, including nitrogen (N), phosphorus (P) and iron (Fe). Some of these elements are retained in the fish body to build new biomass, which acts as a stored reservoir of nutrients, while the rest is excreted or egested, providing a recycling flux to water. Fishing activity has modified the fish biomass distribution worldwide and consequently may have altered fish-mediated nutrient cycling, but this possibility remains
largely unassessed, mainly due to the difficulty of estimating global fish biomass and metabolic rates. Here we quantify the role of commercially-targeted marine fish between 10g and 100kg ($CTF_{10g}^{100kg}$) in the cycling of N, P and Fe in the global ocean, and its change due to fishing activity, by using a global size-spectrum model of marine fish populations calibrated to observations of fish catches. Our results show that the amount of nutrients stored in the global pristine $CTF_{10g}^{100kg}$ biomass was generally small compared to the ambient surface nutrient concentrations but significant in the nutrient-poor regions of the
world: the North Atlantic for P, the oligotrophic gyres for N and the High Nutrient Low Chlorophyll (HNLC) regions for Fe. Similarly, the rate of nutrient removed from the ocean through fishing is globally small compared to the inputs, but can be important locally especially for Fe in the equatorial Pacific and along the western margin of South America and Africa. This model allowed us to compute the spatial distribution of the cycling of elements by the $CTF_{10g}^{100kg}$ biomass at pristine and global peak catch state, which is relatively small compared to the estimated primary production demand for nutrients and estimated
export production of nutrients. Pristine cycling (excretion + egestion) accounted for less than 2.7% of the primary productivity demand for N, P and Fe globally. Relative to the export of nutrients, modeled global pristine $CTF_{10g}^{100kg}$ egestion represents on average 2.3%, 3.0% and 1.1-22% for N, P and Fe (low-high estimates), respectively, with a higher fraction in the low-export oligotrophic tropical gyres. Our study highlights the role of the $CTF_{10g}^{100kg}$ fraction of the icthyosphere (i.e. does not include non-commercial species such as mesopelagic fish) on nutrient storage and cycling, and the potential role of fishing activities on
this cycling, which could be of importance in regions of low nutrient concentration, high fish biomass and/or high productivity demand, and especially at the more local scale for Fe.



## 1   Introduction

Plankton and bacteria dominate the cycling of nutrients in the ocean (Sarmiento and Gruber, 2006) but an increasing number

of studies recognize the contribution of animals to biogeochemical cycles (e.g., Saba et al., 2021). Locally, it has been shown that marine animals can significantly impact the supply and storage of nutrients with consequences for primary production (Cavan et al., 2019; Roman et al., 2014), and interact with the cycling of elements through direct and indirect pathways (Vanni, 2002; Atkinson et al., 2017; Allgeier et al., 2017). For instance, Leroux and Schmitz (2015) described a theoretical framework in which animals control the flux of nutrients up the trophic chain through predation and release of waste products and also

affect the cycling of nutrients through non-consumptive effects (e.g. prey selection and stress induced in preys). In addition, since animals can swim and move in the water column, they are also able to transport nutrients from one place to another, over distances that increase with animal size (Hall et al., 2007; Vanni, 2002; Roman et al., 2014). But thus far there has been little effort to estimate how the global fish population, which we term "ichtyosphere", influences large-scale nutrient cycling.

During their lifecycle, fish assimilate, store and recycle essential elements that they need to build their body tissues. This

storage of nutrients within fish biomass is important for human nutrition as wild-caught fish globally provide essential proteins and other micronutrients (Hicks et al., 2019). Apart from a direct interest for humans, the accumulation of nutrients in fish tissues could be significant for primary productivity compared to nutrients otherwise available dissolved in the water since nutrients stored in fish biomass are not available for primary producers. As an example of this competition for resources, Hjerne and Hansson (2002) showed that fish may compete with primary producers for P in the Baltic Sea, when their biomass

increases. In contrast to the accumulation of nutrients in biomass, the cycling of elements by fish may act as a source of nutrients to primary producers. On one hand, fish recycle elements through the excretion of dissolved bioavailable components that may support part of the primary production. Cycling of N and P by fish has often been studied in freshwater systems but little is known for the global ocean (Schindler and Eby, 1997; Vanni, 2002; Vanni et al., 2006; Griffiths, 2006). For instance, McIntyre et al. (2008) showed that fish are able to create hotspots of recycled nutrients in streams that could meet more than

75% of the algae and microbes requirement for N. On the other hand, fish egest particulate products that can be mineralized and enhance productivity or that can sink to depth and export elements (Davison et al., 2013; Saba and Steinberg, 2012), so that they are no longer available for primary producers. If the stoichiometry of egested particles differs from the stoichiometry of their food, this can also modify the relative availability of nutrients through the water column (Le Mézo and Galbraith, 2020).

The amount of nutrient stored and cycled by fish can vary with different environmental and physiological factors, in space

and time (e.g., Halvorson and Small, 2016; Prabhu et al., 2016; Francis and Côté, 2018; Czamanski et al., 2011; Allgeier et al., 2014). In addition to natural variations, anthropogenic activities, mostly fishing, modify the storage and cycling of nutrients by the icthyosphere. For instance, Layman et al. (2011) and Allgeier et al. (2016) analyzed the cycling of N and P by fish in fished and un-fished coastal sites of the Bahamas and the Caribbean, respectively. Layman et al. (2011) showed lower recycling rates of nutrients by fish in fished sites, due to biomass reduction and habitat fragmentation. Beyond fish biomass reduction, Allgeier

et al. (2016) stressed the role of community size structure that, influenced by fishing, also led to reduction in nutrient storage and cycling.



Although numerous works have identified significant local effects, little is known about the contribution of the icthyosphere to nutrient budgets at the global scale. Maranger et al. (2008) used global fish catch data to estimate the total removal of N by commercial fisheries. They integrated a spatial component in their analysis by computing N removal in 58 Large Marine

Ecosystems (LMEs) and compared it to fertilizers runoff in each off these LMEs. Moreno and Haffa (2014) took a similar approach and estimated the amount of Fe removed each year by fishing from 1950 to 2010.They also used biomass estimates from literature to quantify the amount of Fe in the global fish biomass, and the amount of Fe cycled by this biomass each year.However, the total inventories and cycling rates have remained unquantified due to the lack of reliable global fish biomass and metabolism estimates.

In this paper, we use a model of commercially-targeted marine fish (CTF) to estimate the total CTF biomass and cycling rates and their distribution in the world's oceans. We investigate the amount and spatial distribution of nutrients stored and cycled by the CTF biomass betwen 10g and 100kg, $CTF_{10g}^{100kg}$, in a pristine state and at the global peak catch, as well as the flux of nutrients removed by fisheries at the time of the global peak catch (Fig. 1).

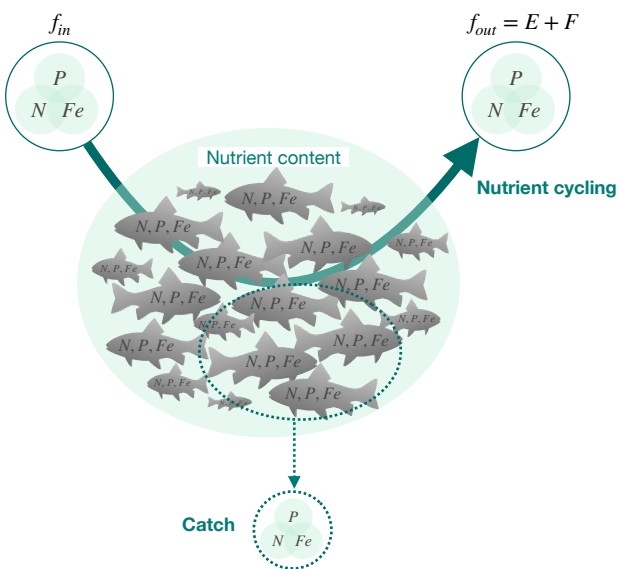

**Figure 1.** Icthyosphere interactions with nutrients. Schematic of the cycling of nutrients (N, P and Fe) by the global fish biomass, the nutrient content of this biomass, and the removal of nutrients from the ocean through fishing. $f_{in}$ and $f_{out}$ are the fluxes in and out of the global fish biomass, respectively. E is for excretion (release of dissolved compounds) and F is for egestion (feces).





## 2 Methods

### 2.1 Model description and simulations


We used an ensemble of simulations following the same method as in Bianchi et al. (in press). We used global fish biomass estimates, under pristine and fished ocean conditions, simulated by the BiOeconomic mArine Trophic Size-spectrum (BOATS) model (Carozza et al., 2017). BOATS represents commercially-targeted marine organisms (here simply called "fish"), larger than 10g and under 100kg, hereby called $CTF_{10g}^{100kg}$, by coupling an ecological and a fishery economics model (Carozza

et al., 2016, 2017). The ecological model is based on processes derived from macro-ecological theory (Carozza 2016). It is parameterized through a Monte Carlo approach, that compares observed and simulated catch in LMEs (Carozza et al. 2017). Fishing effort and catch are computed assuming open-access dynamics and based on the Gordon-Schaefer fishery economics model (Gordon, 1954; Schaefer, 1954) with increasing catchability over time (Galbraith et al., 2017). Galbraith et al. (2019) hypothesizes that fish growth may be affected by Fe scarcity in the wild and demonstrates that the implementation of a simple

form of Fe limitation of fish in BOATS better represents fish catch in HNLC regions. We thus used a version of the model with the Fe limitation of fish growth describe in Galbraith et al. (2019).

The model runs on a 1-degree two-dimensional grid, forced by monthly climatologies of observed net primary production and surface ocean temperature (Carozza et al., 2016). Fish are divided into 3 size groups defined by the asymptotic mass of fish; small (0.3 kg), medium (8.5 kg) and large (100 kg) and, each size-group is divided into 30 mass classes ranging from 10 g

to 100 kg (Carozza et al., 2016). From the Monte Carlo ensemble of simulations, we select 31 (similar to Bianchi et al. (in press)) that best match the historical harvest in LMEs as observed and reconstructed by the Sea Around Us Project (Pauly and Zeller, 2016). Each simulation includes 200 years without catch to estimate pristine biomass at equilibrium, followed by 220 years with fishing driven by the only increase of the catchability of biomass at 7%.y$^{-1}$ to reproduce the historical progression of the global fishery (Bianchi et al., in press). From the 31 simulations, we here analyze the global biomass and cycling rates

at pristine state and at the time of the global peak catch.

### 2.1.1 Nutrient content of fish

The mean nutrient content of the global $CTF_{10g}^{100kg}$ biomass is calculated using mean values of $0.6 \pm 0.2\%$ P and $2.8 \pm 0.4\%$ N in fish wet weight (Table1). We estimate the average body Fe content of fish to be about 21 $\mu$molFe/molC based on whole body Fe measurements (Galbraith et al., 2019; Prabhu et al., 2016), but as the Fe content of fish is poorly constrained (few whole

body measurements) we rather use the 95% confidence-interval, which ranges between 10 and 200 $\mu$molFe/molC (Galbraith et al., 2019). This way we can estimate the effects of the uncertainty on fish body Fe:C.

Body nutrient concentration of fish may be affected by several factors such as body size, ontogeny, speciation, sex, diet, temperature or water nutrient concentration (e.g., Halvorson and Small, 2016; Prabhu et al., 2016; Allgeier et al., 2017). Among these our model could best account for change during ontogeny as organisms grow in size. Yet, analysis of data (see supple-

ment) shows little to no systematic variation of specific nutrient content with size. While other factors can't be represented in our model, some of their effect are included in the uncertainty around the values in table 1. We assumed constant nutrient





**Table 1.** Mean nutrient content in fish, zooplankton and phytoplankton, and mean absorption efficiencies (A) of N, P and Fe for fish used in this study.

| | % N in ww | % P in ww | C/N (molC/molN) | C/P (molC/molP) | Fe:C ($\mu$molFe/molC) |
|---|---|---|---|---|---|
| **Fish** | | | | | |
| Global ± std (range) | [a]2.8± 0.4 | [a]0.6± 0.2 | [a]4.6 (3.4-6.4) | [a]49 (29-82) | [c,e]21 (10-200) |
| **Zooplankton** | | | | | |
| Global± std (range or Fe-poor/Fe-rich, low/high) | [b]1.4± 0.3 | [b]0.14± 0.04 | [a]4.7 (2.9-7.7) | [a]140 (84-231) | [c]30.6 (8.2/85, 4.1/248) |
| **Phytoplankton** | | | | | |
| Global (Fe-poor/Fe-rich, low/high) | From relationships in Galbraith and Martiny (2015) (Eqs. 4) | | | | [f]60.5 ([c]5/92, [f]2.13/258) |

| | C | N | P | Fe |
|---|---|---|---|---|
| **Fish mean absorption efficiency (A)** | [a]0.88 | [a,b]0.86 | [a,b]0.71 | [d]0.24 |

Data from [a]Czamanski et al. (2011) (% converted back to wet weight using a 25% dry weight in wet weight, geometric mean, P absorption efficiency is computed from the linear regression between predator and prey C/P ratio), [b]Schindler and Eby (1997), [c]Galbraith et al. (2019), [d]Thodesen et al. (2001), [e]Prabhu et al. (2016), [f]Moore et al. (2013). "ww" is for wet weight, Fe-poor and Fe-rich refer to the conditions in which the organisms lived, low/high are the low and high estimates from gathered data in Galbraith et al. (2019). Standard deviations are the arithmetic standard deviations, associated with the arithmetic mean. Ratio mean values are geometric means and the ranges are the 95% confidence interval, except for fish Fe:C for which the range is estimated from Galbraith et al. (2019).

proportions throughout food webs and focused on the removal by fishing (Fig. 1).

Although the model non-explicit represents all organisms between 10g and 100kg, including molluscs and crustaceans, 105 hereby called "fish", we used the nutrient content values of fish as they represent the largest proportion of the commercial catch. This may result in an slight overestimation of N and P content, and an underestimation of the Fe content of the modeled fish biomass, part of which is included in the uncertainties we computed (Table 2 and Supp. Table A1).

### 2.1.2 Nutrient cycling by fish

First, we define the excretion and egestion terms considering a single fish. Figure 2a represents the fate of a nutrient element 110 when ingested by a fish. A fish $i$ assimilates part of the nutrients it ingests, $AI_i$, with $I_i$ the amount ingested, while the rest is egested in the form of fecal pellets, $F_i = (1 - A)I_i$ (Fig.2). The absorption efficiency, $A$, is defined as the proportion of an element that goes across the gut of fish, i.e. ingestion net of egestion. Published estimates of absorption efficiencies of fish are listed in Table 1. Part of the elements absorbed across the fish gut are then used for growth and reproduction, $Pr_i = \alpha AI_i$, where $\alpha$ is the somatic assimilation efficiency. The trophic efficiency, $te$, is commonly defined as the amount of an element that 115 is integrated into new biomass following ingestion, and is equal to $\alpha A$. The remainder is used for maintenance and excreted back to the water; $E_i = (1 - \alpha)AI_i$ (Fig. 2a,b). In our model ensemble, $te$ has an average value of 17% for wet biomass, which

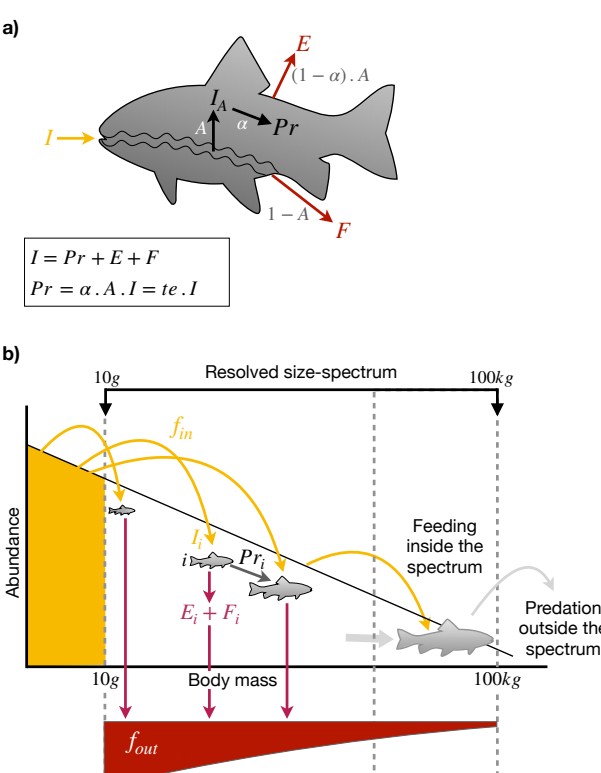

**Figure 2.** Schematic of the flow of elements a) within an individual and b) through the size-spectrum. $I$ = ingestion, $Pr$ = production, $I_A$ = part of ingestion that is assimilated, $E$ = excretion, $F$ = egestion (feces), $A$ = assimilation efficiency net of egestion i.e. absorption efficiency, $\alpha$ = somatic assimilation efficiency, i.e part assimilated allocated to growth, $f_{in}$ = flux entering the spectrum, $f_{out}$ = flux going out of the spectrum

is directly proportionnal to carbon content (Bianchi et al., in press).

Second, at the community level we estimate the total flux of elements by computing the input flux through all feeding on unresolved prey, i.e. "outside" the spectrum (Fig. 2b). By doing so, we avoid accounting for internal cycling of nutrients within the spectrum. We use the model mean predator-prey mass ratio (PPMR), about $6.0 \ 10^3$ in our ensemble, to select all the fish in the spectra whose size allows them to feed only on organisms smaller than the smallest fish of the spectrum (10g). We also assume that the flux entering the spectrum from predation on larger unresolved organisms is compensated by a related external mortality, i.e. by large predators not resolved here such as marine mammals or birds. For the sum of all fish feeding outside the resolved spectrum, the outgoing flux of an element, $f_{out}$, is given by the flux entering the resolved spectrum, $f_{in}$, minus the production of new biomass, $Pr$:

$$f_{out} = \sum_i (I_i - Pr_i) = \sum_i (E_i + F_i) = f_{in} - Pr \tag{1}$$





This flux is estimated from BOATS turnover of biomass. We used the turnover rates of carbon through the fish population, $f_{in}(C)$, the absorption efficiencies of a nutrient X, $A_X$, and the nutrient to carbon ratio within the prey, $(X:C)_{prey}$, to compute the cycling of nutrients through the fish biomass as:

$$f_{in}(X) = f_{in}(C)(X:C)_{prey} \quad ; \quad E(X) = \sum_i (A_X I_i(X) - Pr_i(X)) \quad ; \quad F(X) = f_{in}(X)(1 - A_X) \quad (2)$$

with $f_{in}(X)$ being the flux of nutrient X through the fish biomass, E(X) and F(X) the excreted and egested flux of nutrient X, respectively, for the total fish biomass.

From the computed flux of biomass entering the spectrum for each nutrient, $f_{in}(X)$, we separate the ingested fraction I and egested fraction E using a constant mean absorption coefficient for each nutrient, $A_X$ (Table 1). Note that BOATS provides
ingestion, defined as production divided by local trophic efficiency, $te$, all other quantities are estimated from it. For N and P, we use the percentage of nutrient in wet weight, $p_X$, found in the literature (Table 1), and compute the flux of nutrient as: $f_{in}(X) = p_X f_{in}(ww)$, where $f_{in}(ww)$ is the flux of wet matter through the fish biomass.

To simplify, we assumed that the feeding of fish on organisms outside the spectrum occurs mostly on zooplankton and we used mean zooplankton N and P nutrient content to compute the cycling of these elements through the fish biomass (Table 1).
Zooplankton Fe content appears to vary significantly between Fe-rich and Fe-poor regions (Table 1, Galbraith et al. (2019)). In order to be thorough, we computed Fe cycling in three different ways based on the various computation of the Fe:C distribution of zooplankton: 1) we used a mean Fe:C in zooplankton of 30.6 $\mu$molFe/molC, 2) we used a spatial variation between Fe-rich and Fe-poor regions and used the Fe:C mean values in Fe-poor and Fe-rich regions, respectively, from Table 1, 3) we used the same spatial variation, but with the low and high Fe:C estimates of zooplankton from Table 1. For the spatial variations
of Fe:C, we assumed that Fe-poor conditions are encountered in HNLC regions, which are determined by a concentration of surface nitrate $[NO_3^-]$ larger than 5 mmolN/m3. In order to take into account the gradient between these regions, we locally weighted zooplankton Fe content using a Michaelis-Menten function:

$$(Fe:C)_{zoo} = (Fe:C)_{zoo}^{Fe-rich\,or\,max} + ((Fe:C)_{zoo}^{Fe-poor\,or\,low} - (Fe:C)_{zoo}^{Fe-rich\,or\,high})\frac{[NO_3^-]_{surf}}{5 + [NO_3^-]_{surf}} \quad (3)$$

### 2.1.3 Primary producers demand, nutrient concentrations and export and atmospheric deposition

In our analysis, we compare the nutrient cycling by fish to the nutrient demand by primary producers. We use an averaged satellite-based primary productivity (PP) (Dunne et al., 2007) to compute the PP demand for N, P and Fe. We predict the C:P and C:N ratios in phytoplankton using empirical relationships with $PO_4^{3-}$ and $NO_3^-$ surface concentrations as described in Galbraith and Martiny (2015):

$$(N:C)_{phyto} = 12.5\% + 3\%\frac{[NO_3^-]_{surf}}{0.32 + [NO_3^-]_{surf}} \quad ; \quad (P:C)_{phyto} = 0.6\% + 0.69\%[PO_4^{3-}]_{surf} \quad (4)$$

with nutrient concentrations in $\mu$mol/L. Similarly to zooplankton, the Fe:C of phytoplankton is computed by allowing variation in stoichiometric ratios between the mean value found in Fe-poor conditions and the mean value found in Fe-rich conditions,





or between the high and low phytoplankton Fe:C estimates (Table 1), using a Michaelis-Menten equation:

$$(Fe:C)_{phy} = (Fe:C)_{phy}^{Fe-rich\,or\,high} + ((Fe:C)_{phy}^{Fe-poor\,or\,low} - (Fe:C)_{phy}^{Fe-rich\,or\,high})\frac{[NO_3^-]_{surf}}{5+[NO_3^-]_{surf}} \qquad (5)$$

We did not use a global mean value for phytoplankton, as previous studies and data has clearly showed that phytoplankton
Fe:C values varies significantly between Fe-poor and Fe-rich areas (Table 1).

We use the phytoplankton nutrient ratios to compute the export of nutrients from a satellite-based estimate of total C export (Dunne et al., 2007).

We use the World Ocean Atlas observed $PO_4^{3-}$ and $NO_3^-$ water concentrations (Locarnini et al., 2010), and dissolved Fe concentrations simulated by the biogeochemical model TOPAZ2 (Dunne et al., 2013) to compare the fish biomass nutrient
content to the surface ocean ambient concentrations of nutrients, and to compute the stoichiometric ratios in equation 4 and 5. The TOPAZ2 model represents the cycles of different elements from carbon to calcite and Fe with 30 different tracers and the dynamics of three groups of phytoplankton. Surface concentrations are computed using the 2002-2019 annual mean euphotic depth from MODIS-Aqua (2018). Compared with the single estimation from surface by MODIS, $[PO_4^{3-}]$ and $[NO_3^-]$ observations are discretized along the water column, with 32 levels with a 10m interval or more. Computation of coarse $[PO_4^{3-}]$
and $[NO_3^-]$ over the euphotic layer leads to rough edges of the nutrient concentration maps (Fig. 3b).

Finally, we use current atmospheric deposition fields of soluble N (Brahney et al., 2015) and Fe (Mahowald et al., 2009) (Supp. Fig. A1, ref [1]) to compare to the amount of nutrients removed at the time of global peak catch.

## 3   Fish biomass: a living pool of nutrients

Our results show that the quantity of nutrients contained within the $CTF_{10g}^{100kg}$ biomass represents a non-negligible proportion
of the ambient dissolved concentrations of the respective nutrients in areas where these concentrations in seawater are low and/or where the $CTF_{10g}^{100kg}$ biomass is high.

The highest amounts of N, P and Fe in the pristine fish biomass are located in the most productive regions along the coasts, where most simulated fish biomass occurs (Fig. 3a). Globally, the estimated pristine biomass of $CTF_{10g}^{100kg}$, which represents $2.5 \pm 0.8$ Gtons of wet biomass, contains $69 \pm 31$ Tg of N, $15 \pm 14$ Tg of P and $0.012 - 0.23$ Tg of Fe (Table SA1), of which
about half is found in the Large Marine Ecosystems (LMEs) (Table 2).

The N content of the $CTF_{10g}^{100kg}$ biomass appears more significant compared to ambient surface $NO_3^-$ concentrations, up to more than 50%, in the oligotrophic gyres where nutrient concentrations are low, and in coastal areas where large fish biomass accumulates in the pristine ocean (Fig. 3b), thus explaining the high percentage on average over the LMEs (Table 2). The amount of P in the $CTF_{10g}^{100kg}$ biomass represents a high proportion of available $PO_4^{3-}$ in the North Atlantic ocean, which is
relatively P-poor, up to more than 30 % in some areas (Fig. 3b). The ratio also reaches values larger than 20% in the western North Pacific and in a few locations such as in the Arabian sea. $CTF_{10g}^{100kg}$ biomass stores much higher Fe compared to dissolved surface concentrations in the subarctic North Pacific, subarctic North Atlantic, along 40°S and in coastal areas with

---

[1]http://www.geo.cornell.edu/eas/PeoplePlaces/Faculty/mahowald/





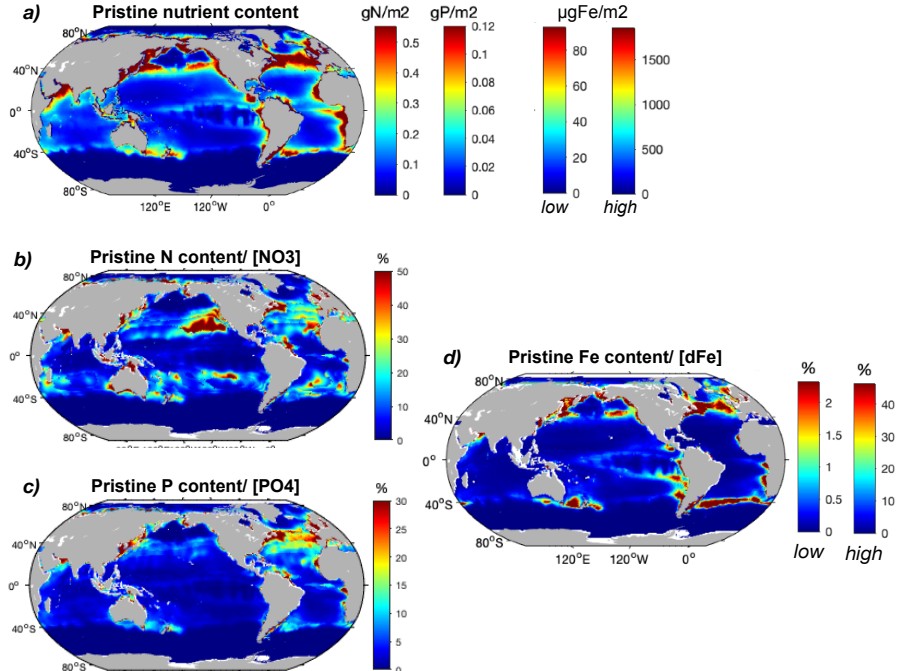

**Figure 3.** Modeled commercial pristine fish biomass mean nutrient content and relative to surface nutrient concentrations. a) P content (mmolP/m2), N content (mmolN/m2) and Fe content (low and high estimates, $\mu$molFe/m2) of the global pristine $CTF_{10g}^{100kg}$ biomass and, b) N content relative to surface $NO_3^-$ concentrations (%), c) P content relative to surface $PO_4^{3-}$ concentrations (%), and d) Fe content (low and high estimates) relative to modeled surface dissolved Fe (%) from the TOPAZ model (Dunne et al., 2013). All surface concentrations are integrated over the 2002-2019 annual mean euphotic depth (MODIS-Aqua, 2018).

relative values as high as 20% for the low estimate (Fig. 3d). The ratio for Fe is particularly low in the Southern Ocean due to the low modeled fish biomass (Galbraith et al., 2019), and in the tropical Atlantic where the input of Fe from dust is the largest

(Mahowald et al., 2009; Myriokefalitakis et al., 2016).

Both low ambient dissolved nutrient concentrations and high fish biomass, as well as a combination of both, result in a significant storage of nutrients by the $CTF_{10g}^{100kg}$ biomass compared to nutrient availability. Thus, in the areas where these conditions occurs, $CTF_{10g}^{100kg}$ biomass has the potential to act as a significant source (if stored nutrients are made available) or sink (if the nutrients cannot be used by primary producers) of nutrients.

## 195 3.1 Comparison to previous estimates

The differences between our estimates and previous studies can be explained by large differences in the biomass of fish, which is difficult to accurately get at the global scale, and, especially for Fe, by the uncertainty on fish nutrient contents on which we lack measurements.





**Table 2.** Table of values in LMEs from the model ensemble simulations in the pristine state and at the global peak catch. This table contains integrated values of: **1.** nutrient content in $CTF_{10g}^{100kg}$ biomass (Tg), **2.** the ratio of nutrient content in $CTF_{10g}^{100kg}$ biomass with surface nutrient concentrations (%), **3.** the amount of nutrient cycled by the $CTF_{10g}^{100kg}$ biomass (Tg/yr), **4.** the ratio of this cycling with the global primary producers demand for these nutrients (%), **5.** the amount of nutrient egested by the $CTF_{10g}^{100kg}$ biomass (Tg/yr) and **6.** its ratio with the exported nutrient quantities (%), **7.** the amount of nutrients removed by fishing in LMEs (Tg/yr) and **8.** the ranges of values of the global nutrients inputs to the ocean from the literature (Tg/yr).

| | N | P | Fe (low estimate) | Fe (high estimate) |
|---|---|---|---|---|
| **1. Content (Tg)** | | | | |
| *Pristine* | $37.4 \pm 17.2$ | $8.1 \pm 7.5$ | $(6.3 \pm 2.1)10^{-3}$ | $0.13 \pm 0.04$ |
| *At global peak catch* | $9.8 \pm 5.4$ | $2.2 \pm 2.2$ | $(1.6 \pm 0.7)10^{-3}$ | $(3.3 \pm 1.4)10^{-2}$ |
| **2. Content/Surface concentration (%)** | | | | |
| *Pristine* | $73.6 \pm 33.8$ | $24.3 \pm 22.5$ | $1.4 \pm 0.5$ | $27.6 \pm 8.7$ |
| *At global peak catch* | $19.1 \pm 10.5$ | $5.4 \pm 5.4$ | $0.29 \pm 0.13$ | $5.7 \pm 2.4$ |
| | | | | |
| **3. Total Cycling (Tg/yr)** | | | | |
| *Pristine* | $101 \pm 55$ | $7.5 \pm 3.9$ | $0.06 \pm 0.12$ | $0.36 \pm 0.09$ |
| *At global peak catch* | $56 \pm 35$ | $4.2 \pm 2.5$ | $0.03 \pm 0.07$ | $0.21 \pm 0.07$ |
| **4. Cycling/PP demand (%)** | | | | |
| *Pristine* | $2.2 \pm 1.2$ | $1.2 \pm 0.6$ | $0.26 \pm 0.53$ | $4.0 \pm 1.0$ |
| *At global peak catch* | $1.5 \pm 0.9$ | $0.85 \pm 0.5$ | $0.15 \pm 0.33$ | $2.6 \pm 0.8$ |
| | | | | |
| **5. Egestion (Tg/yr)** | | | | |
| *Pristine* | $14.1 \pm 12.3$ | $2.2 \pm 1.1$ | $(4.4 \pm 9.1)10^{-2}$ | $(2.7 \pm 0.7)10^{-1}$ |
| *At global peak catch* | $7.8 \pm 4.9$ | $1.2 \pm 0.7$ | $(2.4 \pm 5.2)10^{-2}$ | $(1.6 \pm 0.5)10^{-1}$ |
| **6. Egestion/Export (%)** | | | | |
| *Pristine* | $2.1 \pm 1.1$ | $2.6 \pm 1.3$ | $1.1 \pm 2.3$ | $20.1 \pm 5.0$ |
| *At global peak catch* | $1.6 \pm 1.0$ | $2.1 \pm 1.2$ | $0.8 \pm 1.7$ | $15.7 \pm 5.0$ |
| | | | | |
| **7. Catch (Tg/yr)** | | | | |
| *LMEs* | $2.8 \pm 0.4$ | $0.6 \pm 0.2$ | $(4.7)10^{-4}$ | $(9.3)10^{-3}$ |

| **8. Global inputs to the ocean (Tg/yr)** | N | P | Fe |
|---|---|---|---|
| *Soluble depostion* | $16 - 63^{a,c,e,h,j}$ | $0.1 - 0.5^{a,b,c,d,i,j}$ | $0.6 - 13.4^{b,c,f,g,k}$ |
| *Rivers* | $80^{h}$ | $0.93 - 48^{i}$ | $0.08 - 0.09^{k}$ |
| *N$_2$ fixation* | $140^{h}$ | - | - |
| *Iceberg melting* | - | - | $0.09 - 0.1^{k}$ |

[a] Brahney et al. (2015), [b] Mahowald et al. (2009), [c] Okin et al. (2011), [d] Myriokefalitakis et al. (2016), [e] Fowler et al. (2013), [f] Ito (2015) , [g] Wang et al. (2015), [h] Gruber and Galloway (2008), [i] Benitez-Nelson (2000) , [j] Kanakidou et al. (2012), [k] Moreno and Haffa (2014)





The amount of N stored within the global fish biomass has been previously estimated to be about 23 Tg (Allgeier et al., 2017)[2],

which is about 66% less than our global pristine estimate of $68.7 \pm 30.5$ TgN (Table SA1). Their computation is based on an

estimation of fish biomass of 0.9 Gtons (Jennings et al., 2008) while our ensemble pristine $CTF_{10g}^{100kg}$ biomass is $2.5 \pm 0.8$

Gtons, which is a difference of 64% discussed by Bianchi et al. (in press). Additionally, we used a mean N content of 2.8%,

slightly higher than their value of 2.6%, because we used measurements made on wild fish only and did not try to account

for all the catch diversity in organisms. We only added the species uncertainty around the mean value we use, $\pm 0.4\%$, which

covers the value used in Maranger et al. (2008), to our calculations in Tables 2 and A1.

Moreno and Haffa (2014) estimated that the global fish biomass stored between 0.07 and 0.7 Tg of Fe, which is a slightly

higher range than ours of 0.012-0.23 Tg of Fe (Table 1). To compute these values, Moreno and Haffa (2014) used estimated

fish biomass of 0.9-2 Gtons, which is lower than our modeled estimate of 2.5 Gtons for commercially-targeted fish only due

to a conservative maximum estimation (Wilson et al., 2009). However, they used a range of Fe:C values, 0.073-0.324 gFe per

kg of wet weight (ww) for ray-finned fish, about 3-12 times larger to our 10-200 $\mu$molFe/molC range, equivalent to 0.006-0.12

gFe/kgww (assuming 12.5% C in ww). We are more confident in our compilation of Fe:C values, which is updated, more

complete, and only based on peer-reviewed studies (Galbraith et al., 2019), however the differences highlighted here and the

large range of estimates (low-high values) stress out the high uncertainty on the Fe content of whole fish and the lack of

measurements in this domain.

Note that our modeled estimates are likely to be more reliable in LMEs since fish biomass in these regions is better constrained

by fish catch data. An additional table with the global estimates is provided in the supplementary material (Supp. Table A1).

The new biomass estimate this model provides explains the large difference with previous calculations of the N content of fish,

biomass estimates that are discussed in details in Bianchi et al. (in press). We did not find previous work on the P content of

the global fish biomass and its distribution in the World's oceans. As for Fe, in addition to biomass, the high uncertainty on the

Fe:C content of $CTF_{10g}^{100kg}$ strongly impact the estimations of the total Fe content.

## 3.2   Nutrient content variations in fish and limitations of our study

Many factors are responsible for variations of fish body nutrient concentrations (Table 1), which we can't all model but include

to the best of our abilities in the uncertainties around the mean values we use.

Among the different influencing factors, fish species can be of importance, with for instance bony fish having larger quantities

of P in their body compared to other fishes(El-Sabaawi et al., 2016). Variations in body nutrient contents can also be explained

by sex, e.g. decreased Fe body burden in female rainbow trout during sexual maturation (Shearer, 1984), and variations in the

storage components between organisms (e.g., Czamanski et al., 2011) such as the number and size of bones of vertebrates,

which generally increases with size (Sterner and Elser, 2002). Studies also showed that ontogeny may affect nutrient content

with for example juveniles having less P than adults (Pilati and Vanni, 2007) and the Fe content varying through the life cycle

of salmon (Shearer et al., 1994).

---

[2]It seems that there is a typo in Allgeier et al. (2017). Their estimate is based on Jennings et al. (2008)'s wet biomass estimate of 9 10[8] tons of fish and
they used a 2.6% of N in fish, thus the total N harvest should be 23.4 Tg and not 233.4 Tg





Our model only integrates size as a differentiating factor between fish, so we analyzed aquatic animals body nutrient and body size data from Vanni et al. (2017) for any existing relationship between size and nutrient content. We found no relationship between body N content and size, and only a weak but significant relationship between body P and size (Supp. Fig. C1). In this data set, the changes of P content with size seem to be more related to the difference between vertebrates and invertebrates,

and to be significant for benthic organisms more than pelagic organisms. A recent study as indeed showed that the taxonomic identity is prominent in driving nutrient content variations compared to size (Allgeier et al., 2020). In addition, Hjerne and Hansson (2002) found no significant changes in the N and P content of fish with species (sprat and herring), fish size, seasons or different areas of the Baltic Sea, as did Griffiths (2006) for the P content of different fish species in lakes. Available data for Fe, does not allow us to draw conclusions on the variations in Fe content with size. Our model does not include distinctions

between fish species or diet so we did not test the effect of theses factors on nutrient storage but included the uncertainty on the mean nutrient content of fish in our calculations of the total uncertainty (Tables 1, 2 and Supp. Table A1). Consequently, we did not include variations of body nutrient content with size, the only factor included in our model, in our calculations but simply used the mean values in Table 1 and the uncertainties around these values.

## 4   Nutrient cycling by the commercial fish biomass

Nutrients excreted by fish can directly be used by phytoplankton and bacteria as they often are in a bioavailable form (e.g., Vanni et al., 2006; McIntyre et al., 2008) while egested material tends to sink rapidly (Wotton and Malmqvist, 2001). Because they are capable of movement and alter the stoichiometry of particles, fish can have different impacts than grazing. We gauge these potential impacts by estimating the rates at which fish cycle nutrients, what fraction of primary productivity (PP) this cycling represents, and how much it can contribute to the export of nutrients from the euphotic zone as sinking egested materials. Note

that this is to illustrate the potential magnitude, which could be built only upon with coupled fish biogeochemistry modeling.

The global cycling, i.e. excretion plus egestion, of nutrients by the pristine $CTF_{10g}^{100kg}$ biomass represents about $210 \pm 113$ Tg of N per year, $15.6 \pm 8.0$ Tg of P per year and 0.12-0.77 Tg of Fe per year, of which about half is in the LMEs (Table 2 and Supp. Table A1). Like nutrient storage, modeled cycling by fish is larger where the biomass is higher (Fig. 4 and Supp. Fig. D1). The three different ways of computing Fe cycling by the commercial fish biomass show similar spatial patterns, but

larger Fe cycling when using the weighted spatial variation between the low and high Fe:C estimates in zooplankton (Fig. 4, Table 1). The spatially weighted computations strongly reflect the possibility that Fe cycling by $CTF_{10g}^{100kg}$ might be reduced in HNLC regions (principally in the Southern Ocean and the subarctic Pacific Ocean).

### 4.1   Nutrient cycling by commercial fish and primary producers demand

Our analysis shows that if the nutrients are readily available, fish only represent a small source of recycled elements, larger in

the regions where primary producers (PP) demand and fish biomass are high and in areas where the concentration of nutrients is low. Indeed, he modeled N cycling by pristine $CTF_{10g}^{100kg}$ biomass contributes on average to $2.2 \pm 1.2\%$ of the N demand of PP in LMEs (Table 2), and globally accounts for less than 5% of the demand, except in some coastal areas where it can be as





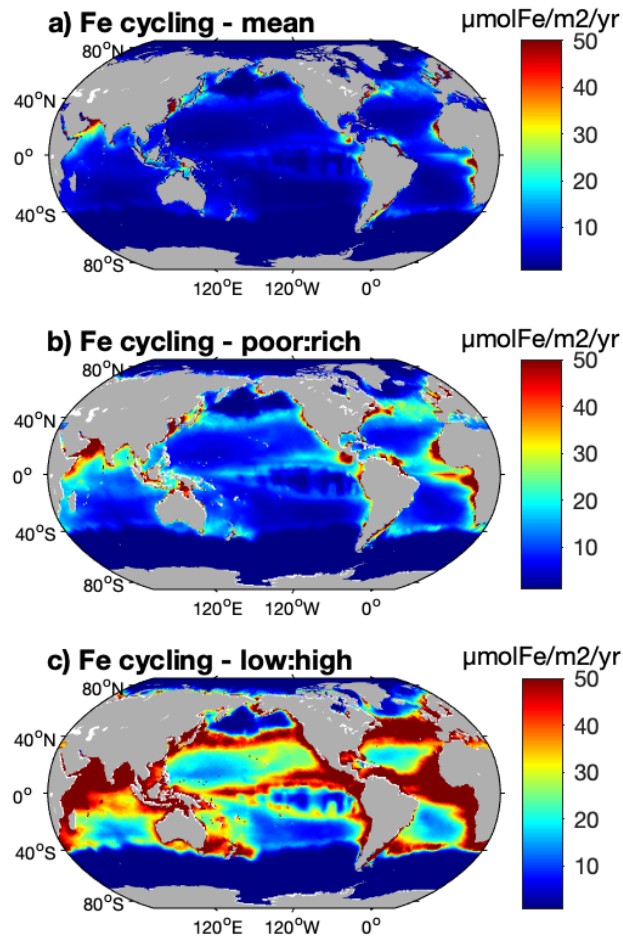

**Figure 4.** Iron cycling by the $CTF_{10g}^{100kg}$ biomass computed using a) a mean Fe:C in zooplankton, b) a spatial variation between zooplankton Fe:C mean values in Fe-poor and Fe-rich areas and based on [NO3] concentrations, and c) the same spatial variation but using the low and high estimates of zooplankton Fe:C from data. (For more details see Methods)

high as 14% (Fig. 5a). Overall, the modeled pristine P cycling accounts for less than 4% of the global P demand, with a larger contribution in the North and equatorial Atlantic coastal regions, and larger than 6% contributions in some coastal areas (Fig.

5b). This P cycling by $CTF_{10g}^{100kg}$ represents 1.2% of the total P demand in the LMEs (Table 2). The high-end estimate of the Fe cycling by $CTF_{10g}^{100kg}$ relative to PP demand for Fe is slightly more significant than the ratios for N and P, as it represents up to 13% of the PP demand for Fe in some coastal areas, and less than 10% everywhere else (Fig. 5e).

This model framework allowed us to analyze the fish nutrient cycling at the global scale, and our global estimates are broadly consistent with prior local studies. For all the coral reefs in the ocean, Allgeier et al. (2014) estimated that the total fish com-

munity supply about 1.2 Tg of N per year, which is about 0.6% of our global pristine estimate (0.8% compared to global cycling at peak catch), or 1.2% of the LMEs pristine estimate (2.1% of our N cycling by the $CTF_{10g}^{100kg}$ biomass at global





peak catch) (Table 2)). This is consistent with the fact that coral reefs cover about 255000-600000 km$^2$ (Spalding and Grenfell, 1997), which is about 0.4-0.8% of the LMEs area. Hernández-León et al. (2008) estimated that zooplankton supply about 1780 TgN/yr worldwide, representing 12-23% of the requirements of phytoplankton and bacteria. This estimated zooplank-

ton cycling is about 11 times our modeled recycling rate by the pristine $CTF_{10g}^{100kg}$ biomass and 6-12 times what $CTF_{10g}^{100kg}$ cycling could provide for primary producers. The Sheldon spectrum has a slope of 0 for the biomass of all marine organisms (**?**), assuming zooplankton mass ranges from $10^{-12}$g to 10g it already represents 3 times more orders of magnitude compared to $CTF_{10g}^{100kg}$, and likely as much more cycling. In addition, due to their higher metabolic and ingestion rates, zooplankton cycling rates are likely to be much higher than fish cycling rates, also exemplified by our modeled cycling size spectrum which

has a negative slope (Supp. Fig. B2). Consequently, due to their high biomass, high metabolic rates and trophic proximity to primary producers (Maldonado et al., 2016; Griffiths, 2006), small plankton mediates most of the nutrient recycling. McIntyre et al. (2008) showed that fish excretion can be important in supplying N and P to primary producers when conditions of high fish biomass and high PP demand or low ambient nutrient concentrations are combined, and when nutrient inputs from anthropogenic sources are low. Our results indeed suggest an increased contribution of fish cycling when these conditions are

combined. However, the strength of this contribution also depends on the timing between the the release of nutrients by fish and the the productivity needs for these nutrients as we will discuss a little more in the case of nutrient budgets with fishing activity.

As for Fe, the $CTF_{10g}^{100kg}$ cycling represents a more important fraction of the PP demand compared to N and P, but large

uncertainties remain in its computation. At the global scale, Moreno and Haffa (2014) estimated that the amount of Fe excreted by the commercial marine fish biomass ranged between 0.4-1.5 TgFe per year. Our modeled estimate range of 0.12-0.77 TgFe/yr, based on the high and low estimates (Fig. 5e and Supp. Fig.E1), is lower but overlapping. The difference can in part be attributed to the lower Fe:C values we used for zooplankton (Table 1), and highlights the uncertainty on the Fe cycling computation (Fig. 4). In the Southern Ocean, whales have been shown to contribute to a maximum of 0.2-0.3% of the phy-

toplankton demand for Fe in a pre-whaling ecosystem and no more than 0.03-0.04% in a post-whaling ecosystem, making their contribution negligible compared to that of zooplankton (>70% for microzooplankton only) (Maldonado et al., 2016). With a modeled contribution of about 0.05-0.5% of the phytoplankton demand for Fe in the Southern Ocean, modeled pristine $CTF_{10g}^{100kg}$ cycling coherently might be able to sustain a larger part of primary productivity than the current whale population, but still far less than zooplankton as discussed before for N cycling.


Similarly to nutrient contents, some but not all types of variability in cycling rates are accounted for in this study. Excretion varies with body size (Vanni and McIntyre, 2016; Allgeier et al., 2015; Hall et al., 2007; Schindler and Eby, 1997), which is taken into account here through the variations of metabolic and production rates within the size-spectrum in the model (Carozza et al., 2017). This allows nutrient cycling to vary with size in our calculations. The effect of temperature on the energy flow and

thus nutrient cycling is also integrated in the model (Carozza et al., 2017). Even though taxonomy, diet, ontogeny, and body nutrient content also influence fish recycling (e.g., Vanni and McIntyre, 2016; Allgeier et al., 2015; El-Sabaawi et al., 2016;


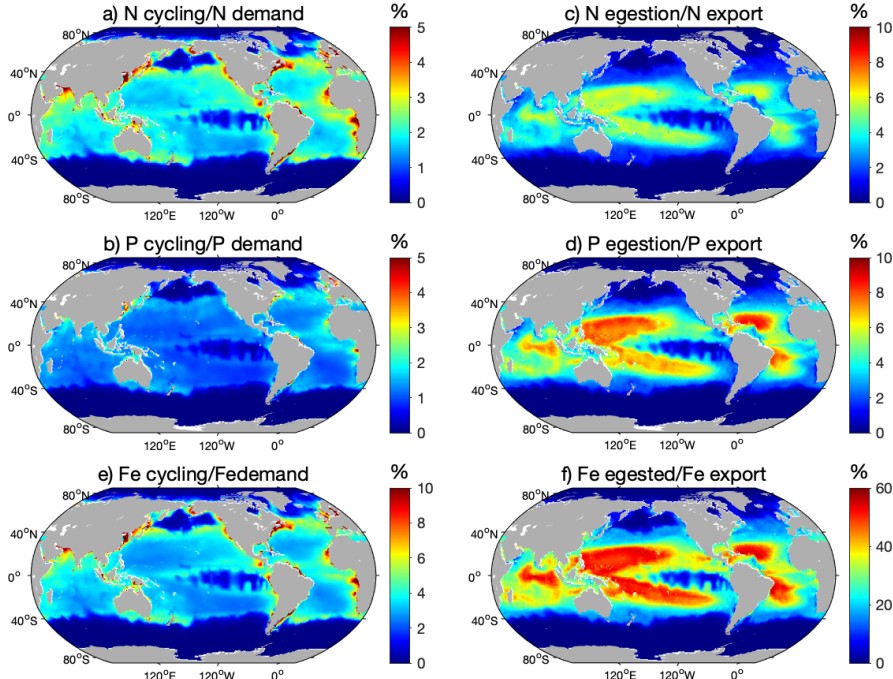

**Figure 5.** Total nutrient cycling and egestion relative to export. Ratio (%) between the amount of nutrients cycled by the modeled pristine global $CTF_{10g}^{100kg}$ biomass and estimated primary producers demand for a) N and b) P, ratios of the modeled amount of nutrient egested by the pristine $CTF_{10g}^{100kg}$ biomass and export at the base of the euphotic zone of c) N and d) P, and high-end estimate of the ratio (%) between the modeled amount of e) Fe cycled by the modeled pristine global $CTF_{10g}^{100kg}$ biomass and estimated primary producers Fe demand, and f) Fe egested by the pristine $CTF_{10g}^{100kg}$ biomass and Fe export at the base of the euphotic zone.The high-end estimates are obtained using Fe cycling computed from the weighted spatial variation between the low and high Fe:C values of zooplankton, and the weighted spatial variation between the averaged Fe:C ratios of phytoplankton in Fe-poor and Fe-rich conditions.

Moody et al., 2015; Pilati and Vanni, 2007; Nugraha et al., 2010) these were not included in our study apart from the uncertainty on zooplankton nutrient content (Table 1). Finally, fish movements also allow the transport of nutrients and constitute a sink of nutrients where the fish forage and a source of nutrients where the fish excrete, egest or die (Vanni et al., 2013; Francis and

Côté, 2018), an effect we do not explicitly include here.

**4.2 Nutrient export by feces**

Our results show that the export of faecal material has the potential to affect the distribution of nutrients within the water column, especially in regions of low export intensity. Egested nutrients are integrated into faecal pellets that sink out of the surface layer and are recycled at larger depths than if bound to smaller particles (Wotton and Malmqvist, 2001; Turner, 2015), espe-

cially fish fecal pellets that can sink faster and deeper than marine snow and phytodetritus (Saba and Steinberg, 2012). Figure 5c-d,f quantifies how much $CTF_{10g}^{100kg}$ egestion may contribute to the export of N, P and Fe to depth using the mean absorption



efficiencies in Table 1 and assuming exported nutrient ratios, without fish, are on average equal to those of phytoplankton. For all the nutrients, $CTF_{10g}^{100kg}$-mediated export accounts for a larger part of the export in the warm, low export regions of the world oceans, i.e. the tropical gyres, where it can contribute up to 50% of the exported Fe for the high-end estimate (Fig. 5f),

6% of the exported N and 10% of the exported P (Fig. 5c,d). Globally, modeled pristine $CTF_{10g}^{100kg}$ biomass egests $29.4 \pm 15.9$ Tg of N, $4.5 \pm 2.3$ Tg of P and 0.009-0.59 Tg of Fe each year, which on average roughly accounts for $2.3 \pm 1.2\%$, $3.0 \pm 1.5\%$ and 1.1-22% of the export of N, P and Fe, respectively, out of the euphotic zone (Supp. Table A1).

These results are in agreement with Davison et al. (2013) who showed that the contribution of mesopelagic fish to the carbon export (via respiration, excretion, egestion and death) is higher in regions where the total export is small. However, Davison

et al. (2013) also showed that locally, in the California Current, the active transport of C by mesopelagic fish alone represents about 15-17% of the total carbon export at depth, with spatial and temporal variations. In their modeling study, Aumont et al. (2018) estimated that, globally, diurnal vertical migration of epipelagic organisms (all migrating fish and zooplankton) contributes to the flux of carbon to depth of about 18% of the passive flux. Egesting and respiring at depth transports significant amounts of carbon and thus also transfers nutrients from the surface to deeper layers, which would probably have increased

the contribution of fish to the export of nutrient in this study if represented in the model.

More than their contribution to total export, fish effect on particles may be most relevant for stoichiometric changes, especially for Fe. Indeed, since the absorption efficiency of Fe is smaller than that of C, the Fe:C in feces will be greater than in the ingested particles (Le Mézo and Galbraith, 2020). Thus, exported fecal material will have a greater Fe:C than biogenic sinking particles made of phytoplankton aggregates or dead organisms. This is potentially important for mesopelagic organisms feed-

ing on sinking material in light of the possible Fe limitation of marine animals (Le Mézo and Galbraith, 2020; Galbraith et al., 2019).

## 5   Extended size-spectrum and total fish biomass

In this study, we only considered the $CTF_{10g}^{100kg}$, which is the target range of the BOATS model, but it does not include all fish in terms of size and non-commercial biomass. First, a more inclusive size range would be from small larvae, about 1g,

to sharks, about $10^6$g, i.e. $CTF_{1g}^{10^6 g}$, which encompasses 6 orders of magnitudes compared to the 4 orders of magnitude of the BOATS size range. We do not include planktonic larvae as small as $10^{-5}g - 10^{-1}g$ because these size classes tend to be dominated by zooplankton (Hatton et al., in press). We first estimate the amount of nutrients stored within this extended marine spectrum, $CTF_{1g}^{10^6 g}$, by analyzing the size spectrum of fish abundance in BOATS. The size spectrum of abundance has a slope of about -1, and the biomass size spectrum a slope of 0 (Fig. SB1) as predicted by the Sheldon spectrum (Hatton et al., in

press). So by extension to smaller and larger sizes, we estimate that the $CTF_{1g}^{10^6 g}$ biomass contains about 1.5 times (6 orders of magnitude versus 4 orders of magnitude) more biomass than the $CTF_{10g}^{100kg}$ biomass, which represents 6.9 Gtons of wet biomass.

These estimates do not take into account non-commercial fish species, especially mesopelagic fish on which there are poor constraints. Bianchi et al. (in press) estimated that the commercially-targeted fish represent only half of the total fish biomass.





Consequently, we could estimate that $CTF_{1g}^{10^6g}$ biomass in the pristine state represents about 20.7 Gtons of wet biomass in this model. The extended size-spectrum estimate of all commercial and non-commercial fish biomass is likely as low as 2-times and high as 8-times the CTF biomass within BOATS' range.

By extending the size spectrum, in addition to increasing fish biomass the icthyosphere would also contains more small fish

than in the standard size range of BOATS, a combination that would change the cycling rates of the total fish biomass. Since smaller fish are shown to have higher cycling rates than large ones, we expect a higher cycling estimates by the global fish biomass. Indeed, in the pristine ocean modeled by BOATS, the C cycling size-spectrum has a slope of $-0.37 molC.m^{-2}.g^{-1}$, and of $-0.57 molC.m^{-2}.g^{-1}$ at the global peak catch (Supp. Fig. B2). Based on these slopes, we would expect that the global fish biomass, including non-commercial fish species, between 1g and $10^6g$, would cycled about 4 times more C in the pristine

ocean and about 3 times more C at global peak catch compared to the $CTF_{10g}^{100kg}$ biomass represented in BOATS in the same conditions.

## 6    Fish catch: anthropogenic extraction of nutrients from the ocean

As fishing activity represents a direct removal of nutrients from the ocean, we estimated how much nutrients were extracted at the global peak catch and how these extraction rates would compare to nutrient inputs to the ocean since our model spatially

represents fish at the global scale. The majority of the catch takes place where fish biomass is modeled to be high: the North Atlantic and Pacific, the east equatorial Pacific, around 40°S and along the coasts. Globally, modeled fishing activity removes $5.4 \pm 0.7$ TgN/yr, $1.2 \pm 0.3$ TgP/yr and $0.09 - 1.8 \ 10^{10}$ gFe/yr from the ocean at the time of global peak catch (Supp. Table A1), of which a little less than 50% in LMEs (Table 2).

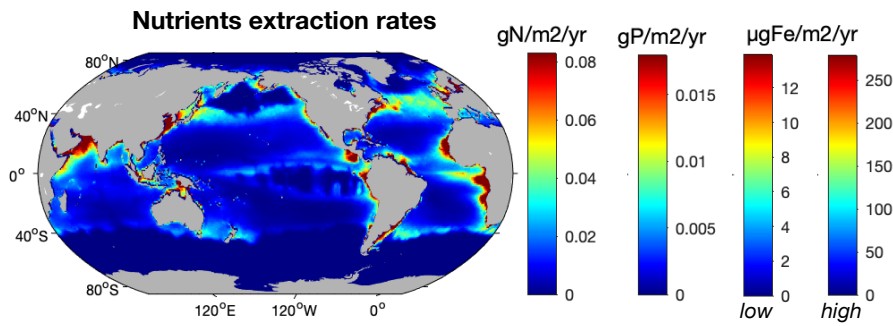

**Figure 6.** Distribution of the modeled amount of N (gN/m2/yr), P (gP/m2/yr) and Fe ($\mu$gFe/m2/yr) extracted from the ocean at the time of the global peak catch.The two colorbars for Fe represent the low and high estimates based on the 95% confidence interval for Fe:C values in fish (10-200$\mu$molFe/molC).



## 6.1 Nitrogen

Our estimate of N extraction from the $CTF_{10g}^{100kg}$ catch is in coherent with previous estimates and the differences mostly reflects the choice of fish biomass estimates in previous studies compared to the one used to calibrate this model. For example, using catch data, Maranger et al. (2008) estimated the amount of N returned to land via fishing in LMEs to be about 0.9 TgN/yr in the 60's and 2.3 TgN/yr in 2000. Our estimate of the N content of fish catch in LMEs is $2.8 \pm 0.4$ Tg of N per year, which is slightly larger than Maranger et al. (2008) 2000's estimate. The difference might be explained by the fact that our model is

calibrated to fit observed catches over the years 1950 to 2010 (Pauly and Zeller, 2016; Bianchi et al., in press) and the catch estimate we use is at the time of the global peak catch. Moreover, our mean N content of fish (2.8% of wet weight) (Table 1) induces a difference of about 0.2 Tg N per year. Allgeier et al. (2017) also estimated the amount of N globally harvested to be about 2.072 Tg of N per year using Maranger et al. (2008)'s 2.6% N content and the FAO catch data. The FAO data only contains reported catches and consequently is lower than the SAUP data used in Maranger et al. (2008) and used to calibrate

BOATS, which explains part of the difference with our estimate of $5.4 \pm 0.7$ TgN/yr at the global scale (see Bianchi et al. (in press) for more details).

Our study framework allows to spatially compare extracted nutrients to nutrient inputs, which was not the case in previous work. For N, even though N extraction by fishing can be significant locally compared to N deposition at the surface, N extraction is negligible compared to the other sources of N to the surface layers. Figure 7a compares the amount of N extracted by fishing

to the modeled soluble atmospheric N deposition from Brahney et al. (2015). Globally, fishing removal of N is smaller than current modeled atmospheric deposition of soluble N, with the higher values, up to more than 60% of the N deposition, in the southern equatorial Pacific, along the western margins of Africa and South America and in the Arabian Sea, where catch is high and deposition is low (Supp. Fig. A1). However, most of N supply to the surface ocean occurs through vertical diffusion and mixing of the upper layers (Sarmiento and Gruber, 2006), which likely accentuates the fact that N extraction by fishing is

insignificant at the global scale.

## 6.2 Phosphorus

Similarly to N, our estimate of P extraction by fishing is coherent with previous work and shows that it is very small compared to inputs of P to the ocean andd resupply from vertical mixing and diffusion in the water column. We estimate that the amount of P removed by fishing at the global scale amounts $1.2 \pm 0.3$ TgP/yr, of which $0.6 \pm 0.2$ TgP/yr occur in the LMEs (Table 2 and

Supp. Table A1). Huang et al. (2020) estimated that wild and aquaculture fisheries, including finfish, crustaceans and molluscs, represented 1.1 Tg of P in 2016, which agrees very well with our estimate. However, we should note that their calculation is based on a catch of 169 Tg of biomass containing finfish, crustaceans and molluscs, while the global amount of catch we modeled is higher, $196.2 \pm 57$ Tg of wet weight, even though Huang et al. (2020) also considers aquaculture in addition to wild captures. Our catch estimate at global peak catch likely overestimate catch (Bianchi et al., in press). Additionally, our

estimate is solely based on fish P content (Table 1), which may overestimate the amount of P extracted by fishing activity since crustaceans and molluscs have lower P content than finfish (Huang et al., 2020).





Contrary to N, the modeled removal of P from harvest would largely exceeds the atmospheric deposition of soluble P as P inputs to the ocean mostly occur through riverine inputs, which represent more than 90% of the total P input to the oceans (Table 2). Consequently, catch transfers P from the ocean to land where as P supply to the ocean is mostly occurring in the coastal areas,

with possible impacts on the P budget of the open ocean (Huang et al., 2020). But similarly to N, vertical diffusion and mixing of the upper layers supplies P to the surface ocean in quantities that most likely render P extraction by fishing insignificant.

## 6.3 Iron

Fe extraction by fishing activity is within the range of previous estimates but large uncertainties remain attached to the estimation of the Fe:C of fish. Moreno and Haffa (2014) investigated the extent to which commercial catch has globally translocated

Fe from the ocean to land. They estimated the global rate of translocation of Fe to be between 0.007 and 0.03 Tg in 2010. Our modeled global range of Fe removal is about $0.0009 - 0.018$ TgFe/yr. Our lower estimated values can once again be explained by the difference in the Fe:C ratios used for fish. Even though our estimates is lower than previous work, it shows that locally Fe extraction can be significant compared to Fe inputs from dust deposition. Indeed, the high-end estimate of Fe extracted is globally small relative to modeled soluble Fe deposition from Mahowald et al. (2009), but it reaches values larger than 100%

in the coastal eastern equatorial Pacific and in some other coastal areas such as Western South Africa, Northern Europe and Canada (Fig. 7b), where modeled Fe deposition is small and harvest is high (Fig. SA1). Contrary to N and P, Fe has a much shorter residence time and thus is subjects to local perturbations, among with Fe extraction by fishing could be important.

## 6.4 Local and time-dependent nutrient budgets

Nutrient budgets are subject to perturbations in space and in time that can modify the relative strength of the nutrient extraction

by fishing activity. Some local nutrient budgets have been investigated to compare the amount of nutrient extracted by fishing to the nutrient loads (e.g. Hjerne and Hansson (2002)). If we were to do similarly budgets, at the global scale, assuming all P inputs come from rivers and atmospheric deposition, which represents 48.5 TgP/yr (Table 2 and SA2), then extracted P represents 2.5% of the inputs globally (1.2% of the inputs in LMEs). For N, global catch represent about 2% of the combined N inputs from atmospheric deposition (49.6 TgN/yr, Table 2), rivers (80 TgN/yr, Table SA2) and $N_2$-fixation (140 TgN/yr,

Table SA2).

Note that fish extracted from a given area may have foraged elsewhere, especially large fish able to undertake long-distance migrations like tuna, salmon or sharks (e.g., Afonso et al., 2017; Gresh et al., 2000). Consequently, the ratios between extracted nutrients and nutrient deposition may be over- or under-estimated, thus over- or under-estimating the role of fishing as a local sink of nutrients (Vanni et al., 2013). In addition, the relative timing of fishing effort along with phytoplankton growth, nutrient

inputs seasonality and residence times may also modify the importance of fishing activity as a sink of nutrients (e.g., Francis and Côté, 2018; Vanni et al., 2006).





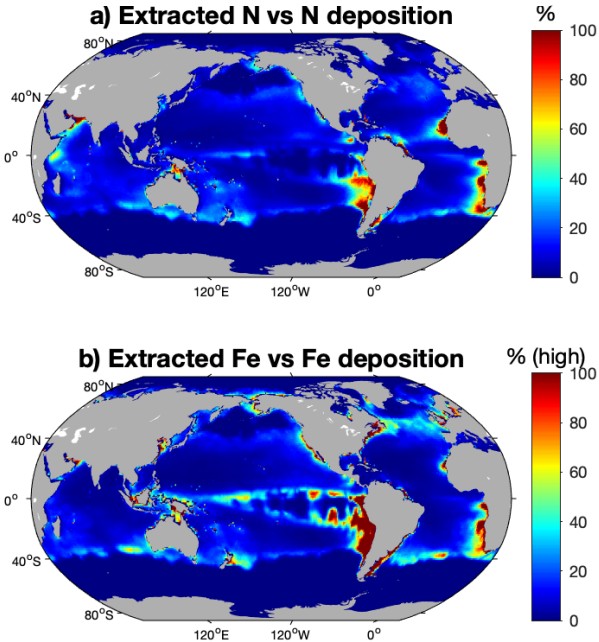

**Figure 7.** Ratio (%) between simulated extracted nutrients and current aeolian soluble nutrients inputs at the surface of the ocean for a) N and b) Fe. "low" and "high" refer to the use of 10 $\mu$molFe/molC or 200 $\mu$molFe/molC in fish, respectively.

## 6.5 Fishing and nutrient cycling

Fishing has a dramatic influence on nutrient cycling, as it removes a lot of biomass, especially in the large size classes, which our analysis clearly shows. In our ensemble of simulations, the cycling rates decrease by about 30% for the three elements

considered at the time of the global peak catch (Table 2 and Supp. Table A1), due to the global reduction in fish biomass of about 60%. The reduction in total nutrient cycling is not as pronounced as the biomass reduction between the pristine state and the global peak catch state because the remaining fish at the global peak catch are smaller and have higher mass-specific metabolic rates (Bianchi et al., in press). By targeting mostly large fish and specific species, fishing modifies the community size structure and trophic interactions thereby changing the animal-mediated nutrient cycling (e.g., Schmitz et al., 2010; Vanni,

2002). In our model ensemble, only changes in the size spectrum structure due to fishing are accounted for (Bianchi et al., in press). Large fish are targeted first, and the size-spectrum of CTF is thus truncated at larger size class (Supp. Fig. B1). This reduction of the mean community size enhances the cycling of elements, because smaller animals tend to have higher metabolic rates. However, the reduction from pristine biomass to biomass at the global peak catch predominates, and globally fish cycling is reduced.



## 445  7   Conclusions

In this study we estimate the amount of nutrients, N, P and Fe, contained in and cycled by the global $CTF_{10g}^{100kg}$ biomass in its pristine state and at the time of the global peak catch. The overall contribution of this commercial icthyosphere is small but significant in regions of low ambient nutrient concentrations, high fish biomass and low export production. Catch represents a small extraction of nutrients globally compared to external inputs, but it removes significant amount of P from the open ocean

compared to external inputs that mainly occurs from rivers. However, N and P cycling by fish is less significant than Fe cycling by fish because N and P are resupplied globally through resuspension by mixing processes, while Fe cycling is much more local and susceptible to perturbations through rapid scavenging for example.

Globally, nutrient cycling by the modeled $CTF_{10g}^{100kg}$ biomass is small compared to primary producers demand for these nutrients. The highest contributions are found close to the coasts where fish biomass and productivity demand are high. Fish

egestion of nutrients via faecal pellets is most important in regions of low export production, i.e. the tropical gyres, especially for Fe. However, fecal pellets may also significantly impact the stoichiometry of sinking particles with consequences for mesopelagic organisms.

Finally, even though these contributions of fish do not appear to be very significant at the global scale they are relevant at the local scale, especially for Fe as mentioned before, and as highlighted by many studies on fish in coral reefs for example.

Fish contributions to nutrient cycles may also be enhanced by top-down effects and trophic cascades that operate on longer time-scales (Kavanagh and Galbraith, 2018). This study does not account for several factors such as fish migrations that would alter the results, especially as in response to warming and deoxygenation due to climate change (e.g., Lefort et al., 2015; Lotze et al., 2019)





**Appendix A: Global values and nutrient inputs to the ocean**

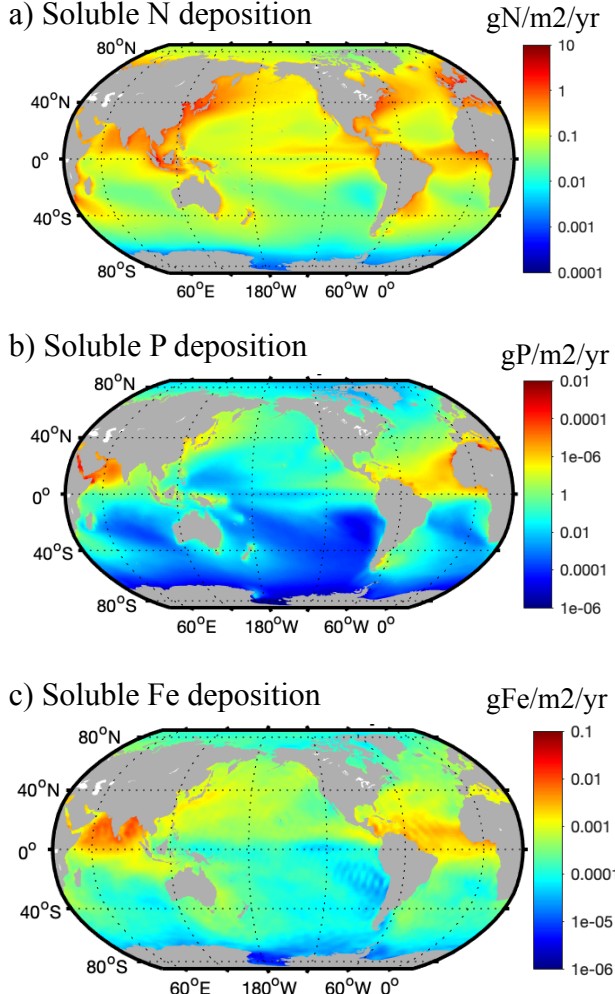

**Figure A1.** Modeled deposition fields of soluble a) N (gN/m2/yr), b) P (gP/m2/yr) and c) Fe (gFe/m2/yr) used to make Figure 7. N and P fields are from Brahney et al. (2015), Fe field is from Mahowald et al. (2009).



**Table A1.** Table of global values from the model ensemble simulations in the pristine state and at the global peak catch. This table contains globally integrated values of: **1.** nutrient content in fish biomass (Tg), **2.** the ratio of nutrient content in fish with surface nutrient conentrations (%), **3.** the amount of nutrient cycled by the fish biomass (Tg/yr), **4.** the ratio of this cycling with the global primary producers demand for these nutrients (%), **5.** the amount of nutrient egested by the fish biomass (Tg/yr) and **6.** its ratio with the exported nutrient quantities (%), and **7.** the amount of nutrients removed via harvest at the global scale (Tg/yr) .

| | N | P | Fe (low estimate) | Fe (high estimate) |
|---|---|---|---|---|
| **1. Content (Tg)** | | | | |
| *Pristine* | $68.7 \pm 30.5$ | $14.9 \pm 13.6$ | $(1.2 \pm 0.4)10^{-2}$ | $0.23 \pm 0.07$ |
| *At global peak catch* | $26.2 \pm 14.7$ | $5.7 \pm 5.8$ | $(4.4 \pm 1.9)10^{-3}$ | $(8.8 \pm 3.8)10^{-2}$ |
| **2. Content/Surface concentration (%)** | | | | |
| *Pristine* | $21.7 \pm 9.6$ | $7.1 \pm 6.5$ | $0.50 \pm 0.16$ | $9.9 \pm 3.1$ |
| *At global peak catch* | $7.9 \pm 4.4$ | $2.3 \pm 2.4$ | $0.17 \pm 0.07$ | $3.4 \pm 1.5$ |
| **3. Total Cycling (Tg/yr)** | | | | |
| *Pristine* | $210 \pm 113$ | $15.6 \pm 8.0$ | $0.12 \pm 0.25$ | $0.77 \pm 0.19$ |
| *At global peak catch* | $145 \pm 88$ | $10.8 \pm 6.3$ | $0.08 \pm 0.18$ | $0.56 \pm 0.17$ |
| **4. Cycling/PP demand (%)** | | | | |
| *Pristine* | $1.5 \pm 0.83$ | $0.91 \pm 0.47$ | $0.16 \pm 0.33$ | $2.7 \pm 0.66$ |
| *At global peak catch* | $1.2 \pm 0.75$ | $0.76 \pm 0.44$ | $0.12 \pm 0.26$ | $2.2 \pm 0.68$ |
| **5. Egestion (Tg/yr)** | | | | |
| *Pristine* | $29.4 \pm 15.9$ | $4.5 \pm 2.3$ | $(9.1 \pm 18.9)10^{-2}$ | $0.59 \pm 0.14$ |
| *At global peak catch* | $20.4 \pm 12.3$ | $3.1 \pm 1.8$ | $(6.3 \pm 13.5)10^{-2}$ | $0.42 \pm 0.13$ |
| **6. Egestion/Export (%)** | | | | |
| *Pristine* | $2.3 \pm 1.2$ | $3.0 \pm 1.5$ | $1.1 \pm 2.3$ | $21.7 \pm 5.3$ |
| *At global peak catch* | $2.1 \pm 1.2$ | $2.7 \pm 1.6$ | $1.0 \pm 2.1$ | $19.5 \pm 6.0$ |
| **7. Catch (Tg/yr)** | | | | |
| *Global* | $5.4 \pm 0.7$ | $1.2 \pm 0.3$ | $9.110^{-4}$ | $1.810^{-2}$ |





**Table A2.** Inputs of nutrients to the ocean.

| Sources (Tg/yr) | N | P | Fe | Reference | Comment |
|---|---|---|---|---|---|
| Atmospheric deposition | 63 | 0.32 | 0.36 | Okin et al. (2011) | |
| | - | 0.17 | - | Myriokefalitakis et al. (2016) | bioavailable P |
| | 29.4 | - | - | Fowler et al. (2013) | |
| | - | - | 13.4 | Ito (2015) | |
| | - | - | 8.4 | Wang et al. (2015) | With Anthropogenic input |
| | 50 | - | - | Gruber and Galloway (2008) | $NO_3^-$ and $NH_4^+$ |
| | - | 0.31 | - | Benitez-Nelson (2000) | pre-anthropogenic, soluble reactive P |
| | 16 (6.4) | 0.35 (0.02) | - | Kanakidou et al. (2012) | organic soluble (anthropogenic contribution) |
| | 36.6 | - | - | Kanakidou et al. (2012) | Total inorganic N |
| | - | 0.24 ( 0.034) | - | Mahowald et al. (2009) | Inorganic P (anthropogenic contribution) |
| | - | - | 0.6-2 | Moreno and Haffa (2014) | |
| Rivers | - | 0.93-4.7 | - | Benitez-Nelson (2000) | pre-anthropogenic |
| | - | 23-48 | - | Benitez-Nelson (2000) | Total with anthropogenic |
| | 80 | - | - | Gruber and Galloway (2008) | |
| | - | - | 0.08-0.09 | Moreno and Haffa (2014) | |
| $N_2$ fixation | 140 | - | - | Gruber and Galloway (2008) | |
| Iceberg melting | - | - | 0.09-0.1 | Moreno and Haffa (2014) | |



## Appendix B: Size-spectrum of abundance, biomass and C cycling

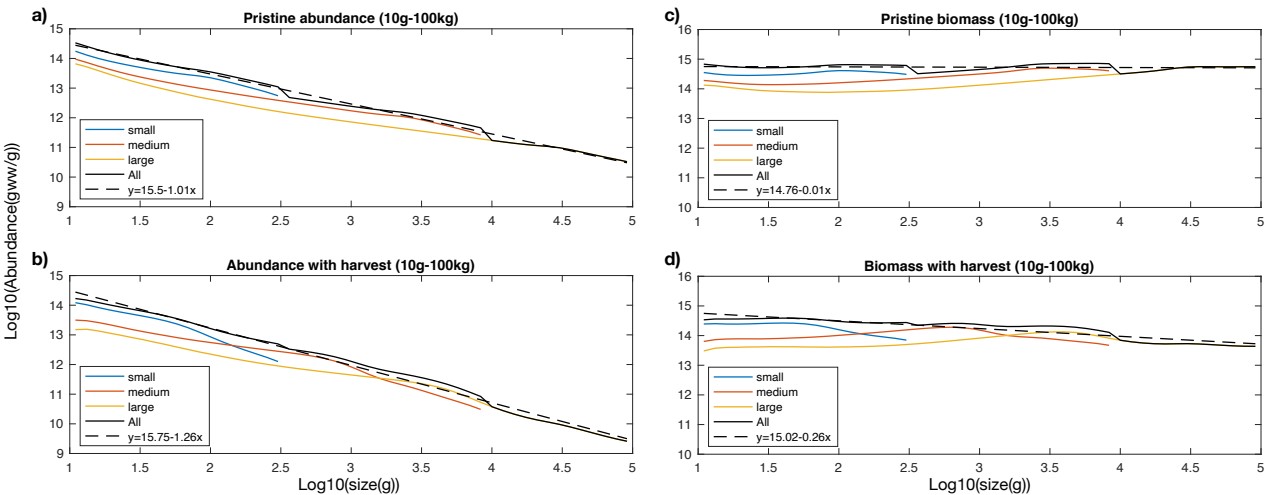

**Figure B1.** Size-spectrum of modeled $CTF_{10g}^{100kg}$ abundance in a) its pristine state and b) at global peak catch, and size-spectrum of modeled $CTF_{10g}^{100kg}$ biomass in c) its pristine state and d) at global peak catch



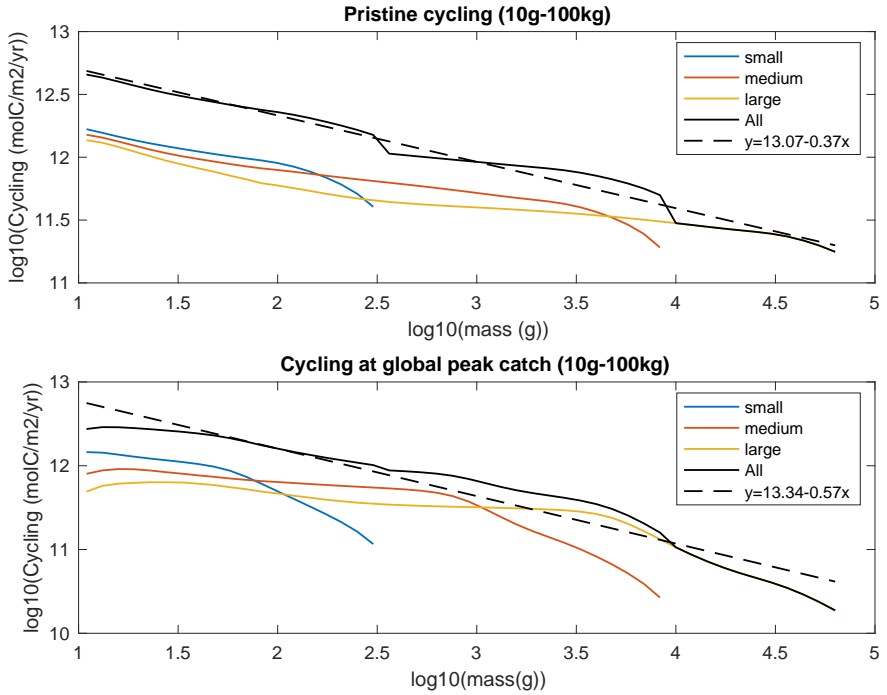

**Figure B2.** Size-spectrum of modeled $CTF_{10g}^{100kg}$ C cycling in a) its pristine state and b) at global peak catch.

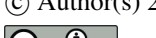



## Appendix C: Body nutrient content versus body size

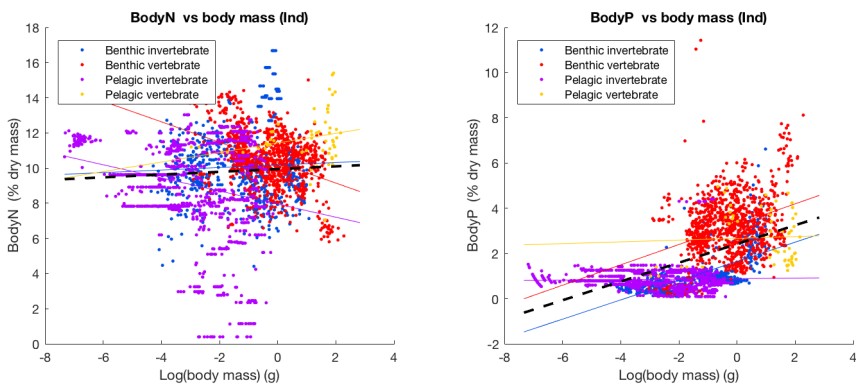

|  | Nitrogen | | | | Phosphorus | | | |
|---|---|---|---|---|---|---|---|---|
|  | Intercept | Slope | R | p-value | Intercept | Slope | R | p-value |
| *All* | 9.9 | 0.08 | 0.07 | 4.3e-5** | 2.4 | 0.42 | 0.58 | 3.4e-299** |
| *Invertebrate* | | | | | | | | |
| All | 9.3 | -0.07 | -0.05 | 0.0138* | 1.3 | 0.12 | 0.34 | 4.2e-63** |
| Pelagic | 8.0 | -0.37 | -0.25 | 6.1e-24** | 0.9 | 0.01 | 0.04 | 0.113 |
| Benthic | 10.2 | 0.07 | 0.05 | 0.207 | 1.7 | 0.43 | 0.66 | 3.3e-87** |
| *Vertebrate* | | | | | | | | |
| All | 10.4 | -0.41 | -0.29 | 2e-21** | 3.2 | 0.37 | 0.29 | 8.4e-22** |
| Pelagic | 11.4 | 0.27 | 0.26 | 0.058 | 2.7 | 0.04 | 0.05 | 0.74 |
| Benthic | 10.3 | -0.58 | -0.39 | 3.6e-37** | 3.3 | 0.45 | 0.34 | 4.0e-28** |

**Figure C1.** Body N (% of dry weight) and body P (% of dry weight), as a function of body mass (log(g)) for pelagic (purple) and benthic (blue) invertebrates and for pelagic (orange) and benthic (red) vertebrates. Regression lines for each type of organisms are shown in the same color. The dashed black line is the global regression line. Regression coefficients are given in the underneath table. Data from Vanni et al. (2017).



**Appendix D: Cycling computation and N and P cycling**

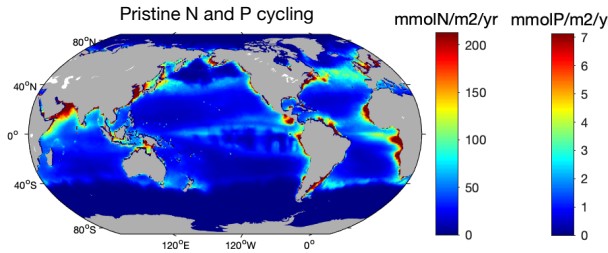

**Figure D1.** N and P pristine cycling.





**Appendix E: Fe cycling versus PP demand for Fe: different computations**



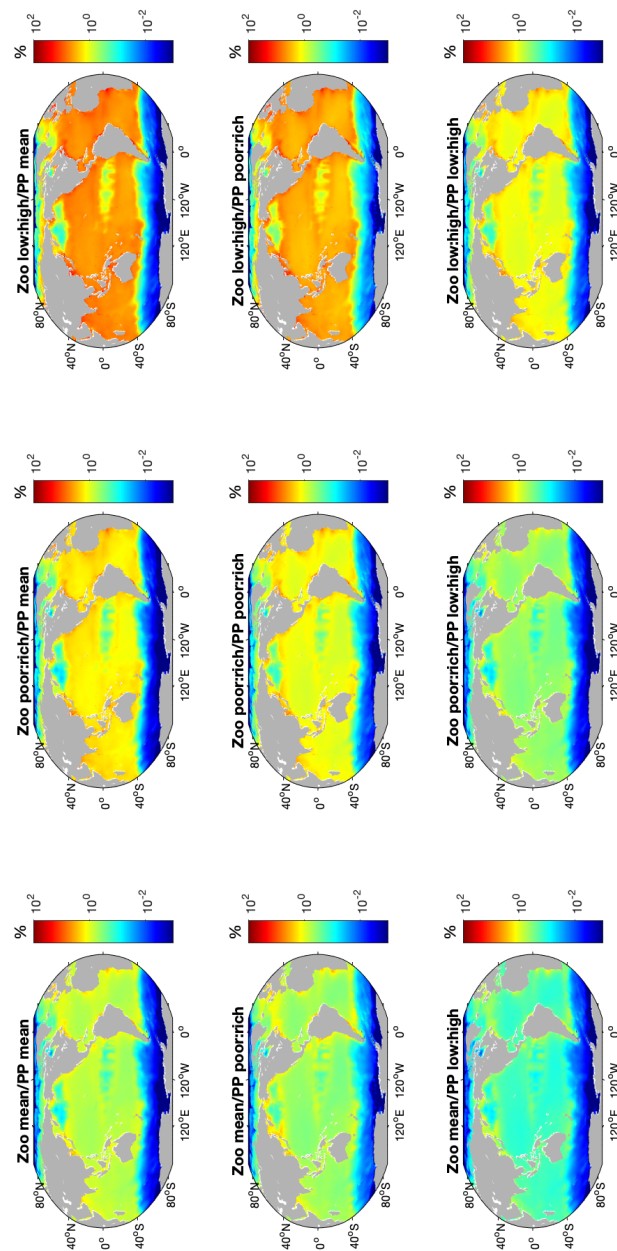

**Figure E1.** Fe cycling divided by phytoplankton demand for Fe. Columns distinguish the Fe cycling computation: left = using a mean Fe:C in zooplankton, middle = using a linear interpolation between the zooplankton mean Fe:C in Fe-rich and in Fe-poor conditions, right = using a linear interpolation between the zooplankton low Fe:C and high Fe:C estimates, interpolations are based on nitrate concentrations as a proxy for HNLC/non-HNLC areas. Lines distinguish the Fe demand of phytoplankton computation: top = linear interpolation between mean Fe:C values in Fe-rich and Fe-poor conditions, bottom = linear interpolation between low and high Fe:C estimates (Table 1).





*Author contributions.* PLM, EDG and DB designed the study. JG performed the simulations. PLM made the figures and wrote the manuscript and all authors revised the manuscript.

*Competing interests.* The authors declare no competing interests

*Acknowledgements.* This project has received funding from the European Research Council (ERC) under the European Union's Horizon 2020 research and innovation programme (grant agreement No 682602). D.B. and J.G. acknowledge support by California Ocean Protec-
tion Council grant C0100400, and NASA grant 80NSSC21K0420. Computational resources were provided by the Extreme Science and Engineering Discovery Environment (XSEDE) through allocation TG-OCE170017.



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
