# Peer review of "Global nutrient cycling by commercially-targeted marine fish"

_Biogeosciences, 2021_

## Author Comment (AC1)

**Authors' responses to reviewers**

We thank both reviewers for their useful and supporting comments on the manuscript. Here is our detailed answer to them.

Reviewer comments are in *italic*, our response is in normal font, changes in the text are highlighted between « citation quotes » and removed text is  while additional text is colored in green.

**Authors' response to reviewer #1 Jacob Allgeier:**

*The study by Le Mezo et al. is an interesting foray into understanding the role of global marine fisheries for biogeochemical cycling. I applaud the authors for providing an interesting perspective on this topic and believe that this study will be a useful contribution to the literature. I have two general comments that I think need to be addressed before being accepted for publication. First, there needs to be some text that clearly tells the reader how this study differentiates from Bianchi et al. As of right now that is not clear. There is obviously massive overlap and I am generally ok with that given all the work done, but the authors simply need to address this up front. Second, the modeling needs to be explained much better. After reading this MS carefully, I still don't fully understand what the main BOATS model does. This is absolutely central to the results and the authors need to lay this out clearly for the reader in a way that is very easy to digest and then continue on with the more specific details. As of right now the methods are not particularly clear in many instances.*

Thank you for your encouraging comments. We agree the methods and model description could be clearer and we modified the manuscript accordingly, as further detailed below.

- *this is a bit confusing – why only egestion? That is a small fraction of excretion+egestion output*
  In this section, we separated the role of excretion and egestion, since we later focus on egestion as it has a distinct role influencing particle stoichiometry (Le Mézo and Galbraith, 2020). We have modified the text to clarify this, see next comment.

- *47-48. this sentence needs some fleshing out.*
  We modified the text as followed:
  « Finally, the egestion of particulate products by fish has been shown to modify the stoichiometry of sinking biogenic particles, including dramatic changes of Fe:C (Le Mézo and Galbraith, 2020), implying that egested material may  also modify the relative availability of nutrients through the water column. »

- *Paragraph at 34. There is an important concept about nutrient capacity that seems missing from this section – the storage of the nutrients by the fish is like the bank account – which matters more or less in systems depending on how otherwise available nutrients are (DeAngelis 1989, Allgeier 2016)*
  We added the bank account analogy and the two references mentioned.
  "During their lifecycle, fish assimilate, store and recycle essential elements that they need to build their body tissues. This storage of nutrients within fish biomass is important for human nutrition as wild-caught fish globally provide essential proteins and other micronutrients (Hicks et al., 2019). Apart from a direct interest for humans,

 the reservoir of nutrients constituted by the fish biomass can matter for the ecosystem, especially phytoplankton, depending on the availability of nutrients otherwise (Allgeier et al., 2016). Because nutrients embedded in fish biomass are not directly available for primary producers, fish can represent  a  competition for resources (Hjerne and Hansson, 2002).  At the same time, the cycling of elements by fish represents a source of nutrients to primary producers as fish recycle elements through the excretion of dissolved bioavailable components . [...] As such, fish biomass can act as a bank account for nutrients, into which nutrients are deposited when fish are feeding rapidly, and withdrawn to the water column when metabolism exceeds predation. "

- *The intro does a great job at covering a very diverse literature base.*
   Thank you.

- *I am increasingly interesting is why 10g is the minimum size of interest. As far as I can tell this would preclude any anchoveta, which is ~ 25% of the worlds catch and definitely plays a bit role in recycling. This seems worthy of at least mentioning at some point.*
   Thanks for pointing out the necessity of clarifying this point. We have added text to clarify this: "The BOATS model was deliberately developed to represent marine organisms over the size range most heavily targeted by fisheries, since this is the range for which fisheries data can offer useful constraints on the ecosystem function (Carozza 2016, 2017). The starting point of 10 g coincides roughly with the weight of mature anchovy (approximately 11 cm in length according to Pauly and Tsukayama (1984)), while above 100 kg the growing significance of  mammals in the ocean makes a strictly ectothermic model less capable of capturing the full marine animal community  (Hatton et al., 2021)."
     - Pauly, D., & Tsukayama, I., 1984, "On the seasonal growth, monthly recruitment and monthly biomass of Peruvian anchoveta (Engraulis ringens) from 1961 to 1979".
     - Hatton, I. A., R. F. Heneghan, Y. M. Bar-On, and E. D. Galbraith. 2021. The global ocean size spectrum from bacteria to whales. Sci. Adv. **7**: 1–9. doi:10.1126/sciadv.abh3732

   We also have a dedicated section (section 5) that discusses how much recycling and storage the fish biomass would represent if interpolated to all fish from small larvae at 1g to sharks at about $10^6$g.

- *The Methods are generally pretty thin – and that is such an important part of what this paper contributes so it seems the authors need to provide a bit more information in that realm throughout.*
   *A few more lines about the model are needed here. Time and again below you lean on this model but never really explain the model to the reader – even in general terms, which I think is needed.*
   *a bit more would be helpful about what is compared*
   Thanks for the suggestion. The Methods section encompasses a description of the model and three subsections describing the computations we made. We agree that the model description could be improved and modified the text accordingly in the Methods section, as well as

adding a new figure to illustrate the basic features of the model: "BOATS represents commercially-targeted marine organisms (here simply called "fish"), larger than 10g and under 100kg, hereby called $CTF_{10g}^{100kg}$, by coupling an ecological and a fishery economics model (Carozza et al., 2016, 2017). The ecological model is based on processes derived from macro-ecological theory (Carozza et al., 2016). It is parameterized through a Monte Carlo approach that compares observed and simulated catch in LMEs (Carozza et al., 2017).

Modeled fish are divided into 3 size groups defined by the asymptotic mass of fish; small (0.3 kg), medium (8.5 kg) and large (100 kg), and each size-group is divided into 30 mass classes ranging from 10 g to 100 kg (Carozza et al., 2016). These groups are not intended to represent the entire marine ecosystem, but rather the sum of all species that have been commercially harvested (and are therefore accounted for in harvest records, which are used to constrain the model). The underlying philosophy of the model is that, although these very diverse species differ widely in their biological strategies, all are competing for food energy ultimately provided by the fixation of organic carbon through photosynthesis (which has been shown to limit fish harvests), while inhabiting the same environment, which therefore makes them subject to the same metabolic constraints. The constraints we apply in the model are the impacts of water temperature on growth, mortality, and phytoplankton size, and the net primary production. Although this biologically 'coarse-grained' approach precludes resolution of species-level dynamics, it is solidly-founded in bioenergetic principles, and is well-suited to the global view of the entire ecosystem on long timescales, given that it is likely to be relatively robust under any changes in the distribution, abundance or evolution of commercial species. Figure 2 provides a schematic overview of the model structure. "

We also added a few lines on how this study compares with that of Bianchi et al., 2020: "We use the same general approach used in Bianchi et al. (2021) to estimate the rate of carbon cycling by the global fish biomass. Our study uses a newer version of the model, and goes beyond the focus on carbon to estimate the role of fish in global nutrient cycling."
and «We thus used a different version of the model from Bianchi et al. (2021) that includes an Fe limitation of fish growth as described in Galbraith et al. (2019). »

- *1- degree two-dimensional grid – explain*
  The model represents fish biomass on a two-dimensional grid, i.e. there is no vertical representation. Biological processes that would in reality be distributed over a vertical range, e.g. primary production, are integrated on the vertical axis. The globe is divided in regular 1° bands in latitude and longitude. We modified the text as:  «The model  represents fish on a two-dimensional grid, i.e. longitude and latitude, which is divided in regular 1° x 1° grid cells. Thus, the model does not resolve the vertical dimension, but sums all ecosystem productivity and biomass within the water column at each horizontal point. Given that the model does not resolve interactions in space between individuals, this reduction in dimensionality does not - on its own - introduce any bias. »

We also corrected the methods subsection numeration as :
«  2.2 Nutrient content of fish »
« 2.3 Nutrient cycling by fish »

«2.1.3 2.4 Primary producers demand, nutrient concentrations, and export and atmospheric deposition »

**Nutrient Content of fish**

- *Allgeier et al. 2021 (supported by Allgeier et al. 2020) showed that body nutrient content is really all about who you are much more than anything else.*
  Thank you for highlighting this useful reference, we have added mention to this study in the text :
  « Body nutrient concentration of fish may be affected by several factors such as body size, ontogeny, species, sex, diet, temperature or water nutrient concentration (e.g., Halvorson and Small, 2016; Prabhu et al., 2016; Allgeier et al., 2017), with species appearing to be the most important factor (Allgeier et al., 2021) ».

- *Table 1. 25% dry wt of wet wt? That seems very high and there is no citation. If I remember correctly, it is typically more like 10% I believe.*
  We choose 25% dry weight of wet weight based on the literature compilation we did in Galbraith et al. (2019, supplementary material). The table is included below. The conversion factor from wet weight to carbon content is generally close to 10%. We added this citation in the table description.

Part of supplementary Table 2 from Galbraith et al. (2019):

| Organism | Dry matter content in wet matter | Ratio dry/wet | Reference |
|---|---|---|---|
| **Fish** | | | |
| Atlantic Salmon | 18-40 % dry matter in ww | 0.18-0.4 | Shearer, 1994 |
| Yellow perch | dw = 22-27.5 % ww | 0.22-0.28 | Hartman & Brandt, 1995 |
| Rainbow trout | dw = 28-38 % ww | 0.28-0.38 | Hartman & Brandt, 1995 |
| Bay anchovy | dw = 11-23 % ww | 0.11-0.23 | Hartman & Brandt, 1995 |
| Anchovies | 25.8-36.9 %dm | 0.26-0.37 | |
| Butterfish | 31 % dm | 0.31 | |
| Capelin | 14.6-22.8 %dm | 0.15-0.23 | |
| Goldfish | 19.4 %dm | 0.19 | |
| Herring | 23.9-32.1 %dm | 0.24-0.32 | |
| Herring, Atlantic | 20.6-28.6 %dm | 0.21-0.29 | |
| Mackerel, Atlantic | 31.2-36.9 %dm | 0.31-0.37 | |
| Mackerel, Pacific | 23.7-33.4 %dm | 0.24-0.33 | Bernard and Allen, 2002 |
| Mackerel, Spanish | 33.8 %dm | 0.34 | |
| Minnows | 18.6 %dm | 0.19 | |
| Salmon | 22.3 %dm | 0.22 | |
| Shrimp, whole | 18.8-23.3 %dm | 0.19-0.23 | |
| Silversides | 26.7-29.3 %dm | 0.27-0.29 | |
| Smelt, ocean | 20.4-25.4 %dm | 0.20-0.25 | |
| Squid | 15.4-18.8 %dm | 0.15-0.19 | |
| Whitebait | 20.4 %dm | 0.20 | |
| Fish | dm=22%wm | 0.22 | Griffiths et al., 2006 |

- *Assuming inverts are the same as fish in terms of nutrients is a very big reach. It really depends on how much of the total biomass they make up, but why not just use invert data instead? It is strange given this point to not make it clear at this point how much of the biomass inverts make up – pretty much either way I think using fish numbers for them is*

*problematic.*

We appreciate this point, however given the high uncertainty of fish biomass (order 40%), the uncertainty in nutrient cycling due to the distinctions among organisms represented by BOATS is actually relatively small. We added a mention on the relative importance of invertebrates vs. fish biomass in the commercial catch (see next comment).

- *Please expand on this as it seems very important.*
  The nutrient contents in invertebrates are slightly larger than in fish for N and P (See Tables 2 and 3 at the end of this document), thus by approximating all commercial catch between 10g and 100kg by fish and thus using fish values we might have underestimated the amounts of N and P within the biomass. However, these approximations are not significant in relation to the uncertainty around the mean values we computed due to the uncertainty of fish biomass.

  We modified the text as follows:
  « Although the model represents all organisms between 10g and 100kg as belonging to three size-spectra of "super-species", including molluscs and crustaceans, we refer to all as  "fish",  and used the nutrient content values of fish as they represent the largest proportion of the commercial catch between 10g and 100kg (from SAUP data in 2018 the ratio of invertebrates to fish in catch is about in between 9 to 30%). The available measurements for nutrient contents of molluscs and crustaceans suggest slightly higher values, but still fall within the uncertainty range around the mean value we used for fish, and we therefore did not attempt to account for invertebrates separately ~~This may result in an slight overestimation of N and P content, and an underestimation of the Fe content of the modeled fish biomass, part of which is included in the uncertainties we computed However, we integrated some uncertainty around these mean values that extend the range of our estimations and reduce the uncertainty induced by this approximation of all commercial catch being fish, especially for Fe2~~1 and Supp. Tables A2-A3). »
  We added two tables to the supplementary material (A2 and A3), which summarize from the literature the measured values of nutrient content of various marine organisms.

- *In general, the use of size spectrum theory needs to be better explained. The reader not familiar with this approach will have a difficult time following and a general overview of the approach is needed. I realize that was the point of Figure 2, but the authors need to go a step further and provide some background as to what size spectrum theory is and why it is appropriate in this application. For example, statements like: "by doing so we avoid accounting for internal nutrient cycling within the spectrum" are not clear. In general, further justification as to why the authors think that the amount coming into the 10g-100kg food web is the same as that leaving needs to be provided.*
  We will add explanations relative to the size-spectrum theory and the use of the BOATS model for this study.
  "by doing so we avoid accounting for internal nutrient cycling within the size spectrum through predation on simulated fish"

- *I think an additional conceptual figure or even table that shows each step of the model is needed for the reader to really be able to grasp and in turn vet the modeling procedures. In addition to being dense, the text is often not very clear and this combination makes understanding the model more challenging than it should be.*
  Included with the more thorough description of the model we have added a new figure to the

paper:

[Figure]

*Figure 2: Schematic overview of the BOATS model. The red, green, and black arrows indicate dependencies of model components on external forcings. The top panel indicates the energetic limits of growth as a function of fish size, while the bottom panel illustrates the three size spectra of fish groups (for simplicity only the large group is represented), their internal dynamics, and link to economics via harvest and the interactive effort.*

- *sentence like this are very common in this MS and add to general lack of clarity. "Export of nutrients" from what and "total C export" from what. There are multiple sentences like this throughout the MS making it difficult for the reader to fully understand what is being done.* Overall we clarified the text of the manuscript.

**Section 2.1.3**

- *The first sentence of this section says that you compare nutrient cycling by fish with the demand of nutrients by phytoplankton. The next sentence says demand was calculated by avg satellite PP, but then the word demand is not mentioned again in this section. There is so much here in this paper and the methods are so extensive, in order for a reader to digest what all was done the writing has to lay it out in simple terms. I would suggest that each sub section has a summary sentence where the whole process is laid out for the reader in simple text and then expanded after that.*
- *"appear more significant compared to.... Up to more than 50%" = this is very difficult text to follow. This whole sentence is a lot to take in. This is another example of the type of text that is throughout the MS that needs to be cleaned up for improved clarity.*

- *text is not clear*

We applied the suggestions here and simplified the text as well as made sure that summary sentences were added clearly to each section.

**Authors' response to reviewer #2 Emma Cavan:**

*Le Mezo et al present a well written and informative article on an important emerging topic, the influence of fishing activities on ocean processes, here specifically nutrient availability and turnover. The study builds on the recent Bianchi et al paper using the same BOATS model but with a focus on nutrient cycling. I think this paper should be published after some revisions to the text outlined below.*

*The large spatial nature of the study means that the authors need to take average nutrient concentrations for fish, which they explain well. They then calculate fish nutrient concentration in an unexploited (pristine) scenario, which is basically a product of fish biomass. I fully support the methods used, but would suggest the authors:*

*1. Explain Fig. 3a in the text clearer. It is not intuitive to have three nutrients represent the same spatial pattern as in Fig. 3a, so explaining again in the results that fish nutrients are held constant and that Fig. 3a is fish biomass distribution would help the reader quickly understand the figure. Especially as in the Introduction the authors recognise that fish nutrients do vary.*

Thank you for the very helpful suggestion, we clarified the text on that point.
We added this sentence in the figure description: "The N, P and Fe content of fish are represented on a single map given that we used a globally-constant nutrient ratio for each element, so that all spatial variation is caused by the fish biomass."
and this in the result section on nutrient content: "Note that in our computation only fish biomass varies spatially as the nutrient content in fish body is held constant (see Methods section)."
We have also modified this paragraph in the Method section : "In this study, we first estimate the quantity of nutrients stored in fish biomass by using a constant nutrient concentration in fish body. Body nutrient concentration of fish may be affected by several factors such as body size, ontogeny, species, sex, diet, temperature or water nutrient concentration (Halvorson et al., 2016, Prabhu et al., 2016, Allgeier et al., 2017), with species appearing to be the most important factor (Allgeier et al., 2021). Among these our model could best account for change during ontogeny as organisms grow in size. Yet, analysis of data (see supplement) shows little to no systematic variation of specific nutrient content with size. We thus assumed constant nutrient proportions throughout food webs."

*2. Either discuss more on the issue of constant nutrients or remove the lines in the abstract (8-13) around Fig. 3b-d, that cycling of certain nutrients in certain regions is important. For instance if P is limiting in the North Atlantic, does that mean that P is reduced in fish biomass? How does the lack of nutrients for phytoplankton propagate up the food chain? Will the fish actually release P in the North Atlantic, or at least enough to have an impact, if they themselves might be P limited? These questions are relevant to the other nutrient limiting regions too. I think if these statements are going to be made upfront in the abstract they should be better supported by discussion on the limitations of the methods*

*and citing other literature, and also including what the main proportion of fish biomass is made up of (i.e. which species) in each region and what the nutrient concentrations might be in those important species groups.*

Thanks for these thoughtful comments. We agree there are many questions that remain in order to fully comprehend how important fish recycling is for different elements, that we cannot fully resolve here due to limited data. Nonetheless, fish show relatively small variations of nutrient content as compared to how much the biomass may vary spatially, so that at the coarse scale, our results are robust. We have attempted to clarify this point better, and have also emphasized the need for further work.

Allgeier et al. (2016) showed that the N and P content of fish depends more on species than anything else, and occurring changes are likely included in the uncertainty ranges we used and computed from data on fish from various regions (Table 1). It is likely that fish with lower requirements for certain nutrients may thrive in regions where these nutrients are scarce.
As for Fe, we lack data for the Fe content of whole body fish in different parts of the Ocean (Galbraith et al., 2019). If fish are less flexible in their stoichiometry compared to plankton then the lower nutrient contents of plankton will have the impact of reducing fish biomass, as we discussed in details for Fe in Galbraith et al. (2019, FiMS). However, if the fish can modify their stoichiometry to match that of their prey then they would have a lower nutrient content. Clearly we lack data for the Fe content of whole body fish. This study highlights that we need more data from fish living in various oceanic regions.

We have added details to the paragraph on the limitations "Nutrient content variations in fish and limitations of our study".

*Focussing the abstract on the main conclusions of the study - total global nutrient cycling with and without fishing - is more inline with what the results can show. I am not convinced showing the spatial patterns is accurate if only biomass changes spatially.*

Here the cycling also changes spatially and the Fe content of plankton as well. Additionally, we think it is interesting to look at spatial patterns when comparing to phytoplankton demand, which also changes spatially. We agree it would be best to have accurate variations of nutrient cycling spatially, however our study has undertaken a more simple approach and the model cannot discretize between species. One could have used catch data to get species and use their associated nutrient content, as did Maranger et al. (2008), to have more accurate nutrient contents, but the induced changes are likely insignificant compared to biomass uncertainty. The model allows us to have estimates of both biomass and nutrient cycling at the global scale, which was our intent.

*In addition, I think it is important to highlight where the limitations are (again spatially) in the BOATS model and algorithms/data used. For instance, the Southern Ocean often does not perform equally well compared to other areas in global algorithms. The biomass will be quite high for the size range of organisms used in some Southern Ocean regions, but is consistently a region of low fish biomass and therefore low fish nutrients according to this study. This should be flagged and acknowledged, or the poles removed from the analysis.*

We agree that it is important to highlight model limitations, and  have added further detail to the model limitations in the Method section (see response to reviewer #1)

As for the Southern Ocean, we used a model version with Fe limitation that strongly reduces the amount of fish biomass modeled in non-coastal regions, improving the model skill in representing fish catches at the resolution of the model (Galbraith et al., 2019, FiMS). We are not aware of studies showing large fish biomasses in the Southern Ocean in the size range of our model (> 10 g), which excludes abundant southern ocean pelagic fish such as myctophids.

*A final point is to discuss what the limitations are of keeping the lower size spectrum (< 10g) constant. There would be feedbacks to the ecosystem of harvesting up the food chain. These include reducing nutrient release which may reduce primary production but also reducing grazing pressures that could influence (increase or decrease) food supply to the resolved size-spectrum. I am not suggesting the authors do this analysis, but I think is a valid point to make given this paper aims to increase our knowledge of all fishery impacts to marine life.*

We entirely agree that the question of what happens to smaller organisms is an important point. We deliberately left aside this portion of the size spectrum in our model framework, given the weakness of observational constraints compared to larger organisms. We added some words on the effects of keeping the lower spectrum constant: "Finally, as the size classes below 10g are kept constant (Fig. 2) we are not able to account for biomass changes that fishing might induce down to plankton, and how it reverberates up to fish through food supply (Dupont et al., in prep), which has the potential to further modify fish-mediated nutrient cycling."

*Text sometimes has stand alone sentences not in paragraph.*

We clarified the text overall.

Table A2 - Fish N and P content

| reference | N content | P content | C content | original units | %N ww | %P ww | C%ww | C/N | C/P | N/P | comment |
|---|---|---|---|---|---|---|---|---|---|---|---|
| Czamanski et al 2011 | 11 | 2,3 | 45,5 | % dry weight | 2,8 | 0,6 | 11,4 | 4,8 | 58,2 | 12,1 | Wild marine fish, Assuming 25% dw in ww |
| | 12,3 | 2 | 46 | | 3,1 | 0,5 | 11,5 | 4,4 | 66,2 | 15,2 | |
| | 12,7 | 2,5 | 45,5 | | 3,2 | 0,6 | 11,4 | 4,1 | 48,1 | 11,5 | |
| | 12,1 | 2 | 46,3 | | 3,0 | 0,5 | 11,6 | 4,5 | 67 | 15,1 | |
| | 9,8 | 2 | 47,5 | | 2,5 | 0,5 | 11,9 | 6,8 | 76,5 | 13 | |
| | 12,2 | 2,2 | 42,3 | | 3,1 | 0,6 | 10,6 | 4 | 54,9 | 13,5 | |
| | 11,6 | 2,3 | 42,4 | | 2,9 | 0,6 | 10,6 | 4,3 | 51,4 | 12 | |
| | 13,4 | 2,9 | 42,9 | | 3,4 | 0,7 | 10,7 | 3,8 | 37 | 10,5 | |
| | 10,5 | 2,9 | 42 | | 2,6 | 0,7 | 10,5 | 4,7 | 42,6 | 8,8 | |
| | 11,5 | 2,4 | 42,8 | | 2,9 | 0,6 | 10,7 | 4,3 | 44,8 | 10,9 | |
| | 7,1 | 3,5 | 43,8 | | 1,8 | 0,9 | 11,0 | 7,3 | 37,5 | 5,1 | |
| | 13 | 3,8 | 41,4 | | 3,3 | 1,0 | 10,4 | 4,5 | 46,4 | 10,7 | |
| | 11,7 | 2 | 46,8 | | 2,9 | 0,5 | 11,7 | 4,6 | 52,7 | 11,8 | |
| | 11 | 3,6 | 38,8 | | 2,8 | 0,9 | 9,7 | 4,1 | 29,1 | 7,1 | |
| | 10,6 | 3,4 | 41,9 | | 2,7 | 0,9 | 10,5 | 4,6 | 35 | 7,6 | |
| Huang et al., 2012 | 10,21 | 2,11 | 45 | % dw | 2,6 | 0,5 | 11,3 | | | | Japanese anchovy,Huanghai Sea, China |
| | 11,26 | 2,45 | | | 2,8 | 0,6 | | | | | Tanner et al., 2000 |
| | 9,7 | 1,49 | 46 | | 2,4 | 0,4 | 11,5 | | | | Sterner and George, 2000 |
| | 10,4 | 3,27 | | | 2,6 | 0,8 | | | | | Dantas and Attayde, 2007 |
| | 10,24 | 2,53 | | | 2,6 | 0,6 | | | | | Dantas and Attayde, 2007 |
| Hjerne and Hansson, 2002 | 2,4 | 0,43 | | % ww | 2,4 | 0,43 | | | | | Herring and sprat, Baltic Sea |
| Beers, 1966 | | | | | | | | 4,57 | 69,93 | 15,31 | Fish and fish larvae, Bermuda, NATL |
| Schindler and Eby, 1997 | 2,54 | 0,5 | | % ww | 2,54 | 0,5 | | | | 5,1 | From Davis and Boyd, 1975; Pencak et al., 1985; Nakashima and Leggett, 1980 |
| Griffths, 2006 | | 2,3 | | %dry weight (22% of ww) | | 0,506 | | | | | |
| | 2,15 | | | | 2,15 | | | | | | see paper |
| | 2,57 | | | | 2,57 | | | | | | see paper |
| | 2,54 | | | | 2,54 | | | | | | see paper |
| | 2,84 | | | | 2,84 | | | | | | see paper |
| | 2,72 | | | | 2,72 | | | | | | see paper |
| | 2,54 | | | | 2,54 | | | | | | see paper |
| | 2,71 | | | | 2,71 | | | | | | see paper |
| | 2,42 | | | | 2,42 | | | | | | see paper |
| | 2,83 | | | | 2,83 | | | | | | see paper |
| | 2,46 | | | | 2,46 | | | | | | see paper |
| | 2,26 | | | | 2,26 | | | | | | see paper |
| | 2,3 | | | | 2,3 | | | | | | see paper |
| | 2,22 | | | | 2,22 | | | | | | see paper |
| | 2,85 | | | | 2,85 | | | | | | see paper |
| | 2,67 | | | | 2,67 | | | | | | see paper |
| | 2,87 | | | | 2,87 | | | | | | see paper |
| Maranger et al., 2008 | 2,59 | | | %ww | 2,59 | | | | | | see paper |
| | 3,2 | | | | 3,2 | | | | | | see paper |
| | 2,7 | | | | 2,7 | | | | | | see paper |
| | 2,36 | | | | 2,36 | | | | | | see paper |
| | 3,02 | | | | 3,02 | | | | | | see paper |
| | 2,88 | | | | 2,88 | | | | | | see paper |
| | 3,02 | | | | 3,02 | | | | | | see paper |
| | 2,66 | | | | 2,66 | | | | | | see paper |
| | 2,9 | | | | 2,9 | | | | | | see paper |
| | 2,85 | | | | 2,85 | | | | | | see paper |
| | 3,26 | | | | 3,26 | | | | | | see paper |
| | 2,38 | | | | 2,38 | | | | | | see paper |
| | 3,49 | | | | 3,49 | | | | | | see paper |
| | 3,41 | | | | 3,41 | | | | | | see paper |
| | 3,18 | | | | 3,18 | | | | | | see paper |
| | 3,2 | | | | 3,2 | | | | | | see paper |
| | 3,79 | | | | 3,79 | | | | | | see paper |
| Ramseyer, 2002 | 2,71 | | | %ww | 2,71 | | | | | | 60 fish species + 6 hybrids |
| Kraft, 1992 | 10 | 2 | | %dw | 2,5 | 0,5 | | | | | From other papers, dry:wet=0.25 |
| MEAN all (arithmetic) | | | | | 2,77 | 0,6 | 11,0 | 4,7 | 51,1 | 10,9 | |
| Standard deviation all | | | | | 0,37 | 0,16 | 0,59 | | | | |
| MEAN Czamanski (arithmetic) | | | | | 2,8 | 0,7 | 10,9 | 4,7 | 49,8 | 11,0 | only marine fish |
| Standard deviation Czamanski | | | | | 0,4 | 0,2 | 0,6 | 1,0 | 13,2 | 2,9 | only marine fish |
| Geometric mean (all) | | | | | 2,7 | 0,6 | 11,0 | 4,6 | 49,3 | 10,4 | |
| Geometric std factor | | | | | 1,14 | 1,32 | 1,05 | 1,18 | 1,30 | 1,33 | |
| Geometric range: min | | | | | 2,4 | 0,5 | 10,5 | 3,9 | 38,0 | 7,8 | |
| Geometric range: max | | | | | 3,1 | 0,8 | 11,5 | 5,5 | 64,0 | 13,8 | |
| geometric CI 0.95 | | | | | 1,3 | 1,7 | 1,1 | 1,4 | 1,7 | 1,7 | |
| 95% min | | | | | 2,1 | 0,4 | 10,0 | 3,4 | 29,6 | 5,9 | |
| 95% max | | | | | 3,6 | 1,0 | 12,0 | 6,4 | 82,1 | 18,1 | |

Table A3 - Invertebrates N and P content

| reference | organism | N content | P content | C content | original units | %N ww | %P ww | C/N | C/P | N/P | comment |
|---|---|---|---|---|---|---|---|---|---|---|---|
| Schindler and Eby, 1997 | copepods | 1,38 | 0,072 | | | 1,38 | 0,072 | | | 19,2 | Andersen and Hessen, 1991 |
| | bosmina | 1,14 | 0,096 | | | 1,14 | 0,096 | | | 11,9 | Andersen and Hessen, 1991 |
| | daphnia | 1,14 | 0,17 | | | 1,14 | 0,17 | | | 6,7 | Andersen and Hessen, 1991 |
| | mysis | 1,72 | 0,18 | | %ww | 1,72 | 0,18 | | | 9,6 | Nakashima and Leggett, 1980 (a,b) ; Davis and boyd, 1975 ; Penczak et al., 1985 |
| | amphipods | 1,72 | 0,18 | | | 1,72 | 0,18 | | | 9,6 | Nakashima and Leggett, 1980 (a,b) ; Davis and boyd, 1975 ; Penczak et al., 1985 |
| Czamanski et al 2011 | microzooplankton | | | | | | | | | 21,5 | Le Borgne (1982) |
| | copepods | | | | | | | 5,63 | 116 | 20,37 | Beers (1996), Ikeda and Mitchell (1982), Uye and Matsuda (1988), Gismervik (1997), Walve and Larsson (1999) |
| | euphausids-mysids | | | | | | | 4,58 | 120,2 | 26,15 | Roger (1978), Beers (1966), Ikeda and Mitchell (1982) |
| | other crustacea | | | | | | | 5,52 | 90,25 | 16,46 | Beers (1966), Walve and Larsson (1999), Gismervik (1997), Pertola et al., (2001) |
| | chaetognaths | | | | | | | 4,16 | 124,99 | 30,06 | Beers (1966), Uye and Matsuda (1988) |
| | salps | | | | | | | 4,39 | 148,99 | 34,14 | Ikeda and Mitchell (1982), Igushi and Ikeda (2004), Le Borgne (1982) |
| | polychaetes | | | | | | | 4,49 | 178,3 | 38,38 | Beers (1966), Ikeda and Mitchell (1982), Clarke (2008) |
| | mollusks | | | | | | | 4,8 | 162,3 | 33,84 | Ikeda and Mitchell (1982) |
| | siphonophores | | | | | | | 4,28 | 200 | 46,72 | Beers (1966) |
| | hydromedusae | | | | | | | 2,91 | 108,7 | 37,4 | Beers (1966) |
| | pteropods | | | | | | | 8,15 | 196,08 | 24,06 | Beers (1966) |
| Griffiths, 2006 | Zooplankton | | 0,92 | | %dw (12%) | | 0,1104 | | | | |
| | Zoobenthos | | 0,8 | | %dm (20%) | | 0,16 | | | | |
| Kraft, 1992 | Zooplankton | 10 | 1 | | %dw | | | | | | From other papers |

| | | | | | | N (%ww) | P (%ww) | C/N | C/P | N/P | |
|---|---|---|---|---|---|---|---|---|---|---|---|
| Arithmetic mean | zooplankton | | | | | 1,4 | 0,14 | 4,9 | 144,6 | 24,1 | |
| Standard deviations | zooplankton | | | | | 0,3 | 0,04 | 1,4 | 38,3 | 11,8 | |
| Geometric mean | | | | | | 1,4 | 0,13 | 4,7 | 140,0 | 25,8 | |
| Standard factor | | | | | | 1,2 | 1,40 | 1,3 | 1,3 | 1,8 | |
| range: min | | | | | | 1,2 | 0,1 | 3,7 | 108,4 | 14,3 | |
| range: max | | | | | | 1,7 | 0,2 | 6,1 | 180,8 | 46,6 | |
| geometric CI 0.95 | | | | | | | | 1,6 | 1,7 | 3,2 | |
| 95% min | | | | | | | | 2,9 | 84,8 | 8,1 | |
| 95% max | | | | | | | | 7,7 | 231,0 | 82,1 | |